# Nanoproteomics enables proteoform-resolved analysis of low-abundance proteins in human serum

Timothy N. Tiambeng[1,5], David S. Roberts [1,5], Kyle A. Brown[1], Yanlong Zhu [2,3], Bifan Chen[1], Zhijie Wu [1], Stanford D. Mitchell[3,4], Tania M. Guardado-Alvarez[1], Song Jin [1✉] & Ying Ge [1,2,3✉]

Top-down mass spectrometry (MS)-based proteomics provides a comprehensive analysis of proteoforms to achieve a proteome-wide understanding of protein functions. However, the MS detection of low-abundance proteins from blood remains an unsolved challenge due to the extraordinary dynamic range of the blood proteome. Here, we develop an integrated nanoproteomics method coupling peptide-functionalized superparamagnetic nanoparticles (NPs) with top-down MS for the enrichment and comprehensive analysis of cardiac troponin I (cTnI), a gold-standard cardiac biomarker, directly from serum. These NPs enable the sensitive enrichment of cTnI (<1 ng/mL) with high specificity and reproducibility, while simultaneously depleting highly abundant proteins such as human serum albumin (>$10^{10}$ more abundant than cTnI). We demonstrate that top-down nanoproteomics can provide high-resolution proteoform-resolved molecular fingerprints of diverse cTnI proteoforms to establish proteoform-pathophysiology relationships. This scalable and reproducible antibody-free strategy can generally enable the proteoform-resolved analysis of low-abundance proteins directly from serum to reveal previously unachievable molecular details.

[1] Department of Chemistry, University of Wisconsin—Madison, Madison, WI 53719, USA. [2] Human Proteomics Program, School of Medicine and Public Health, University of Wisconsin—Madison, Madison, WI 53719, USA. [3] Department of Cell and Regenerative Biology, University of Wisconsin—Madison, Madison, WI 53719, USA. [4] Molecular and Cellular Pharmacology Training Program, University of Wisconsin—Madison, Madison, WI 53719, USA. [5]These authors contributed equally: Timothy N. Tiambeng, David S. Roberts. ✉email: jin@chem.wisc.edu; ying.ge@wisc.edu

The structure, activity, and function of proteins are modulated by posttranslational modifications (PTMs), making the comprehensive analysis of proteoforms[1]—the various protein products arising from myriad PTMs and splicing isoforms from a single gene—crucial for understanding all biological systems at a functional level[2–4]. Mapping the proteoform landscape with absolute molecular specificity represents the next stage of proteomics to achieve proteome-wide characterization and understanding of protein function[3–6]. Top-down MS is the most powerful technology to analyze proteoforms to understand the biological variations that underlie complex phenotypes[3,4,6–8]. However, the MS detection of low-abundance proteins remains an unsolved challenge due to high dynamic range of the human proteome[8–10].

In particular, the human blood proteome, a circulating representation of all pathophysiological processes, is immensely complex and the huge dynamic range of the blood proteome ($10^{12}$) dominated by the presence of numerous highly abundant proteins, such as human serum albumin (HSA), makes the detection of low-abundance proteins extremely difficult[9,10]. Front-end protein enrichment strategies are therefore required to specifically capture and enrich low-abundance proteins from the complex biological milieu prior to MS analysis[9,11]. Antibodies have so far been the dominating affinity reagents for protein capture and quantification in biological research[12,13]. However, antibody-based platforms suffer from significant limitations including the batch-to-batch variability, high cost of the antibody production, and relatively low chemical stability[14,15]. Moreover, the intrinsic challenge of designing and evaluating epitope-specific antibodies for PTMs makes it difficult to develop robust antibodies suitable for proteoform analysis[16]. Hence, there is an urgent need to develop the next generation of affinity platforms capable of global proteoform capture[12,13,17]. Nanoparticles (NPs) are highly effective for such sensitive and specific proteoform enrichment because: (1) they are commensurate in size and diffusion kinetics with proteins allowing effective penetration through complex biological mixtures; (2) they have high surface-to-volume ratios to enhance protein interaction; (3) they are versatile scaffolds to couple diverse affinity ligands for protein binding and capture[18–21].

Cardiac troponin I (cTnI) is a gold-standard biomarker for cardiovascular diseases (the leading cause of death worldwide)[22–25]. cTnI forms the inhibitory subunit of the cTn complex (cTnI–cTnT–TnC; Supplementary Fig. 1) and is released into the bloodstream following cardiac injury[23–25] where it circulates with low abundance (typically < 50 ng/mL)[26] and in myriad proteoforms (e.g., phosphorylated, acetylated, oxidized, and truncated), making detection and analysis extremely challenging. Recent studies have shown that cTnI is heavily modified and its proteoforms arising from various PTMs can provide new insights to the molecular mechanisms underlying cardiovascular diseases[27–31]. The PTM profiles of cTnI can function as molecular fingerprints of cellular signaling pathway activity with the potential to serve as the next-generation biomarkers[27,32,33]. Therefore, a comprehensive proteoform-resolved cTnI analysis that can detect cTnI in blood with detailed PTM information is urgently needed[27].

Here we report an antibody-free top-down nanoproteomics approach enabled by the integration of nanotechnology and top-down proteomics to specifically enrich low-abundance proteins directly from human serum using surface-functionalized NPs and comprehensively analyze the enriched proteoforms by top-down MS. We design an organosilane surface functionalization molecule bearing a cysteine (Cys)-thiol reactive handle for the robust synthesis of peptide-functionalized superparamagnetic NPs to capture and specifically enrich low-abundance proteins from human serum. We focus on the development of a top-down nanoproteomics strategy for comprehensive proteoform-resolved analysis of low-abundance cTnI in the blood. We demonstrate that top-down nanoproteomics can provide a high-resolution proteoform-resolved landscape of diverse cTnI molecular fingerprints arising from various PTMs to establish proteoform–pathophysiology relationships.

## Results

### Synthesis and characterization of peptide-functionalized NPs.

We first rationally designed surface-functionalized superparamagnetic iron-oxide (magnetite, $Fe_3O_4$) NPs to specifically enrich cTnI from complex mixtures (Fig. 1). We synthesized an organosilane linker molecule N-(3-(triethoxysilyl)propyl)buta-2,3-dienamide (Fig. 1a and Supplementary Figs. 2–6), hereinafter referred to as BAPTES, which we use to silanize the oleic acid coated $Fe_3O_4$ NPs[34] following a method for reproducible surface silanization[35]. The terminal allene carboxamide functional group of the BAPTES ligand possesses high chemoselectivity toward the thiol side chain of Cys, and forms a stable and irreversible conjugate not prone to hydrolysis[36]. Importantly, instead of using antibodies to target cTnI, we chose a short, linear peptide evolved for high cTnI affinity by phage display and in silico evolution[37]. The use of peptides offers significant advantages for protein enrichment in comparison with antibodies, such as improved chemical stability to changes in pH and reducing environments, thermal stability, scalability using solid-phase peptide synthesis, and batch-to-batch reproducibility. This specific peptide (HWQIAYNEHQWQ) not only exhibits an impressive binding affinity ($K_d$) of 270 pM comparable to that of antibodies[37], but also targets the central portion of cTnI (amino acid residues 114–144) with less susceptibility to proteolysis and is postulated to be an optimal targeting epitope to detect all forms of cTnI present in blood circulation[38–40]. To evaluate this peptide functionalization approach, we first analyzed the reaction of BAPTES with a C-terminal Cys-modified derivative of the high affinity peptide (HWQIAYNEHQWQC) using high-resolution tandem MS (MS/MS). The peptide-Cys reaction with BAPTES occurred exclusively even in the presence of other biologically relevant nucleophiles, such as hydroxyls, amines, and carboxylates (Supplementary Figs. 7–8). It should be noted that such allene carboxamide chemistry relies on the presence of a Cys nonessential to the peptide binding sequence. In the case where such a Cys is not available, alternative bioorthogonal coupling approaches, such as azides and cyclooctynes, can be employed[41,42].

After confirming such allene carboxamide coupling chemistry, we functionalized the BAPTES silanized NPs (NP-BAPTES) with the high affinity cTnI-binding peptide onto the NP surface (Fig. 1b), which are hereafter referred to as NP-Pep. Transmission electron microscopy (TEM) revealed the uniformity and monodispersity of as-synthesized $Fe_3O_4$-oleic acid NPs with an average diameter of 8.0 ± 0.3 nm (Fig. 1c) and confirmed the morphology did not significantly change after BAPTES silanization (Fig. 1d) and further peptide surface functionalization (Fig. 1e). Physicochemical properties of the surface-functionalized NPs measured at each reaction step confirmed the proper functionalization and elucidated the surface properties of the NPs. Fourier transform infrared (FTIR) spectroscopy analysis of the NP-BAPTES revealed strong peak intensities at 1970 and 1947 $cm^{-1}$ characteristic of the allene-containing molecules (C=C=C)[43] (Fig. 1f). Moreover, such IR signatures of as-synthesized NP-BAPTES were consistent and reproducible across different synthesis batches (Supplementary Fig. 9). High-resolution magic angle spinning nuclear magnetic spectroscopy confirmed the successful silanization of the NPs with BAPTES (Supplementary Fig. 10). After peptide coupling, the relative intensities of the characteristic allene IR peaks were reduced (Fig. 1f), indicating

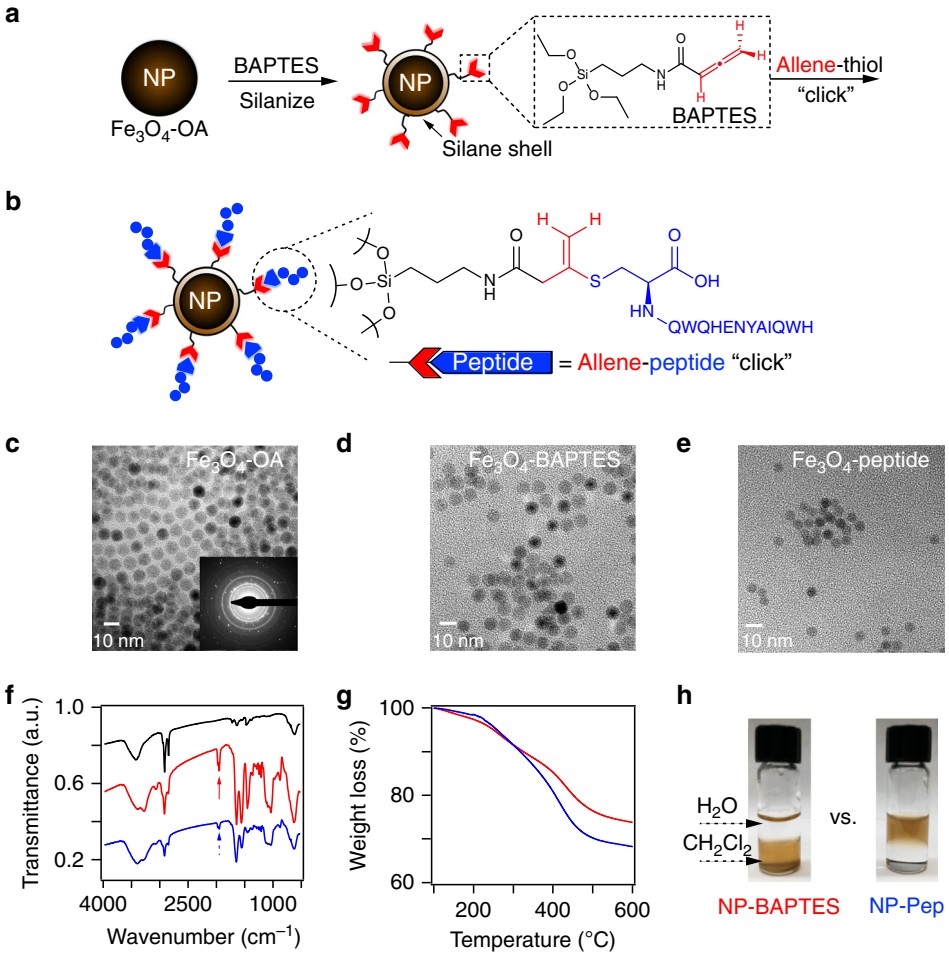

**Fig. 1 Design and characterization of surface-functionalized nanoparticles (NPs) for capturing cTnI. a** Silanization of $Fe_3O_4$ NPs using an allene carboxamide-based organosilane monomer (BAPTES) for cysteine-thiol-specific bioconjugation. **b** Illustration of the rationally designed NPs that are surface functionalized with a 13-mer peptide that has a high affinity for cTnI (NP-Pep) for cTnI enrichment. The 13-mer peptide possesses a C-terminal cysteine that selectively reacts with the allene carboxamide moiety on the silanized NPs. Representative TEM images of surface-functionalized NPs: $Fe_3O_4$-OA NPs (**c**) (inset shows the selected area electron diffraction pattern), $Fe_3O_4$-BAPTES NPs (**d**), and $Fe_3O_4$-Peptide NPs (**e**). Representative FTIR spectra (**f**) and TGA analysis (**g**) of various NPs: $Fe_3O_4$-OA NPs (black), $Fe_3O_4$-BAPTES NPs (red), and $Fe_3O_4$-peptide NPs (blue). FTIR spectra are offset along the y-axis for clarity, with arrows denoting the characteristic allene (C $=$ C $=$ C) vibrational modes (1971 and 1947 cm$^{-1}$) corresponding to the silanized NPs. TGA analysis reveals increased organic content on NPs after peptide coupling. **h** Photographs of functionalized NPs in a biphasic mixture of dichloromethane ($CH_2Cl_2$) and water ($H_2O$), comparing the solvent compatibility of the NP-BAPTES and the NP-Pep. The NP-BAPTES are dispersible in dichloromethane but the NP-Pep are stable and dispersible in water. The displayed NP-Pep and NP-BAPTES originated from the same synthetic batch. NP-BAPTES is referred to as NP-Ctrl in the subsequent figures and text. Data are representative of $n = 3$ independent experiments. Source data are provided as a Source Data file.

the successful consumption of the allene groups upon peptide conjugation. Thermogravimetric analysis (TGA) of the NP-BAPTES revealed a 26% weight loss, accounting for a thin silane coating on the surface-functionalized NPs (Fig. 1g). Comparatively, an increased measured weight loss percentage (32%) was observed after peptide coupling, which suggests the successful attachment of the 13-mer cTnI affinity peptide. From the difference in weight loss (~6%) between the NP-BAPTES and the final NP-Pep, a surface density of ~0.034 µmol peptide/mg NP was inferred (Supplementary Tables 1–3). To demonstrate the colloidal stability of the NP-Pep, we determined the zeta potential (ζ-potential) of the NP-Pep suspended in 0.1× PBS buffer (pH 7.4) to be ~−38 mV, which has been previously shown to be both ideal for serum protein applications and sufficient for electrostatic repulsive forces to dominate over the van der Waals force, such that agglomeration is suppressed (Supplementary Fig. 11)[44,45]. Furthermore, photographs of the NPs dispersed in a biphasic mixture of dichloromethane ($CH_2Cl_2$) and water illustrate the

drastic change in NP-solvent compatibility before and after successful conjugation with the hydrophilic cTnI-binding peptide (Fig. 1h).

**Reproducible and faithful recapitulation of cTnI proteoforms.** We then evaluated the cTnI enrichment performance of the NP-Pep using sarcomeric protein extracts prepared from human heart tissues (Supplementary Fig. 1, detailed conditions shown in Supplementary Table 4), which were found to contain ~0.3% cTnI by enzyme-linked immunosorbent assay (ELISA). We first used sodium dodecyl sulfate polyacrylamide gel electrophoresis (SDS-PAGE) to visualize the protein components in the sarcomeric extracts by SYPRO Ruby staining to confirm the reproducibility of the protein extraction procedure (Supplementary Fig. 12). The cTnI enrichment experiments using the functionalized NP-Pep (Fig. 2a) proceed as following: (1) incubating the NP-Pep in the protein mixture, (2) magnetically isolating the

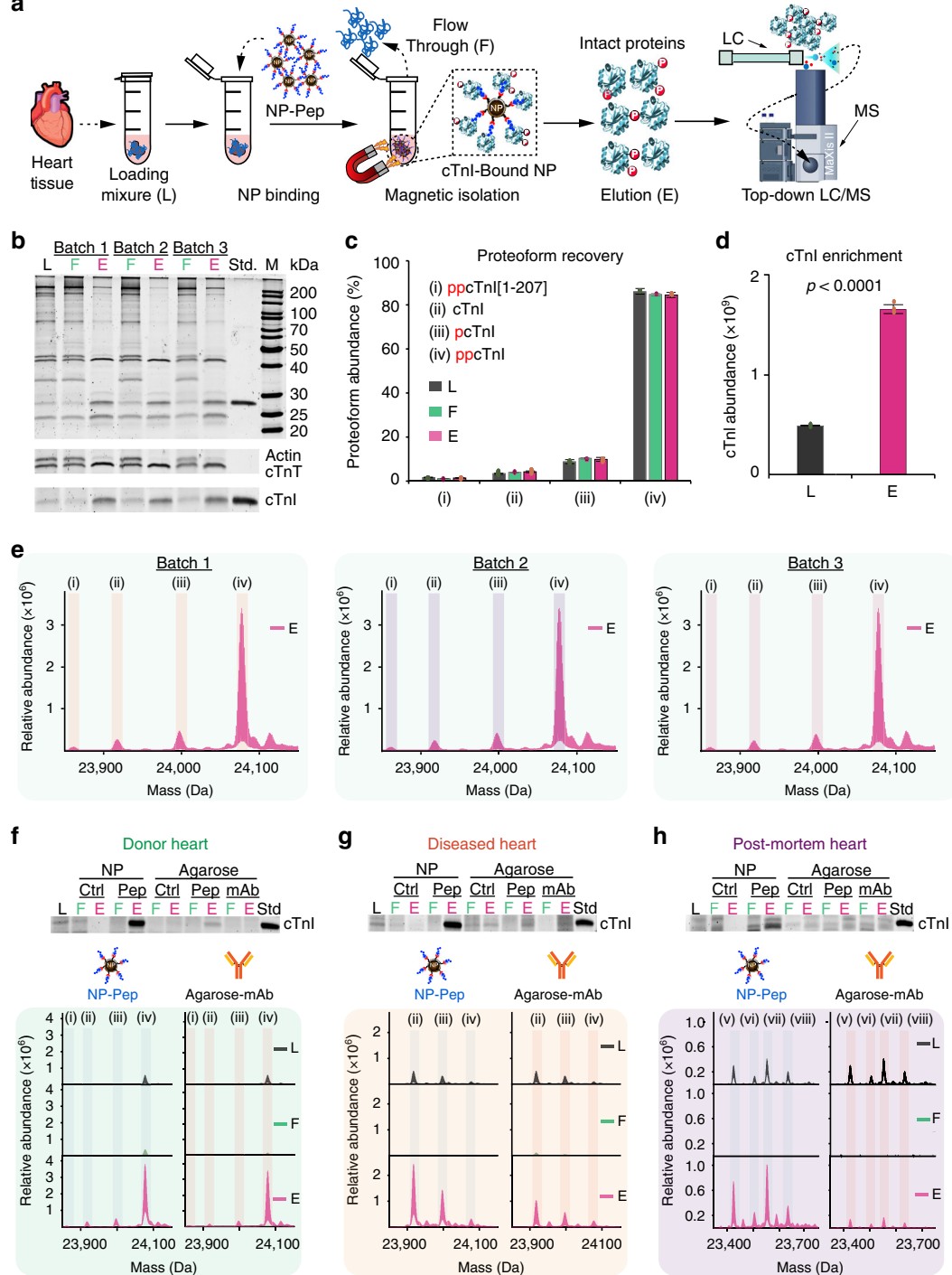

**Fig. 2 Evaluation of enrichment performance and reproducibility of NP-Pep for capturing cTnI. a** Schematic illustration of the nanoproteomics strategy for cTnI enrichment and top-down MS analysis of cTnI proteoforms. Heart tissue extract (loading mixture, L) is first incubated with the NP-Pep. Following magnetic isolation, the nonspecific proteins are removed as flow through (F). The NPs are washed and the NP-bound proteins of interest are eluted (E) and analyzed by top-down LC/MS. **b** SDS-PAGE demonstrates the high reproducibility obtained from three different NP-Pep syntheses (inter-batch). The lower panels highlight the specific enrichment of cTnI and cTnT in E as compared to L and F. Evaluation of total cTnI proteoform recovery (**c**), and relative abundance of cTnI (**d**), before and after NP-Pep enrichment. The same L, F, and E shown in the SDS-PAGE from (**b**) were equally loaded (500 ng) for LC/MS analysis. The deconvoluted top-down mass spectra corresponding to cTnI proteoforms were used to calculate the relative abundance of each cTnI proteoform when normalizing for total protein amount injected. Proteoform abundance data (**c**) are representative of $n = 6$ independent experiments with error bars indicating standard error of the mean. cTnI relative abundance data (**d**) are representative of $n = 3$ independent experiments with error bars indicating standard error of the mean. **e** MS-based evaluation of cTnI enrichment using three different synthetic batches of NP-Pep showing the reproducible enrichment performance. **f–h** Enrichment from three different human heart samples using NP-Pep in comparison with agarose-mAb. Upper panels feature SDS-PAGE strips that visualize cTnI enrichment performance of NP-Pep and agarose-mAb. Equal protein amount (500 ng) of the L, F, and E was loaded on the gel in (**b**, **f–h**). Roman numerals correspond to N-terminally acetylated cTnI proteoforms following Met exclusion: (i) ppcTnI[1-207]; (ii) cTnI; (iii) pcTnI; (iv) ppcTnI; (v) cTnI[1-205]; (vi) pcTnI[1-205]; (vii) cTnI[1-206]; (viii) pcTnI[1-206]. M. protein marker, Std. endogenous cTnI protein standard, p phosphorylation, pp bisphosphorylation. A summary of the enriched proteoforms and their respective mass measurements by top-down MS are listed in Supplementary Tables 5 and 6. Source data are provided as a Source Data file.

NP-Pep to wash and remove unbound nonspecific proteins, and (3) eluting bound cTnI off of the NP-Pep using an acidic buffer to disrupt the intermolecular interactions between NP-Pep and the bound cTnI (see experimental details in the "Methods"). After NP enrichment, the protein bands corresponding to cTnI and cTnT were far more prominent in the elution solutions (E) compared to the initial loading mixtures (L) which contained abundant sarcomeric proteins such as actin (lower panel, Fig. 2b). We found that salt (NaCl) concentration of the wash buffer was a critical parameter and could be tuned to promote effective cTnI enrichment using the NP-Pep (Supplementary Fig. 13). Subsequently, we demonstrated the highly effective and reproducible enrichment of cTnI from sarcomeric extracts by the NP-Pep from both three different intra- and inter-batch syntheses (Fig. 2b and Supplementary Figs. 14 and 15). We further investigated whether the high affinity cTnI-binding peptide functionalization onto the NPs was a critical factor for the high specificity enrichment by using the following NP controls: (1) NP-BAPTES with no peptide, denoted as NP-Ctrl; (2) NP-BAPTES functionalized with peptide at an acidic pH to inhibit formation of the covalent Cys–thiol conjugate; (3) NP-BAPTES functionalized with a negative-control peptide[37] to reduce cTnI-binding affinity. All of these control experiments showed no appreciable cTnI enrichment (Supplementary Fig. 16). This confirms only NPs properly functionalized with the high affinity cTnI-binding peptide (NP-Pep) allow for successful cTnI enrichment.

We next evaluated the cTnI enrichment performance of the NP-Pep by top-down liquid chromatography (LC)/MS analysis of the initial protein loading mixture, the resulting flow through, and the final elution mixtures after cTnI enrichment. Top-down MS provided a bird's eye view of all cTnI proteoforms present for direct quantification of the relative abundance of individual cTnI proteoform normalized to the total cTnI proteoforms[32,46]. Importantly, the NP-Pep preserved all endogenous cTnI proteoform distributions and faithfully retained the endogenous cTnI PTM profiles at every step of the enrichment process with no artifactual modifications (Fig. 2c). Moreover, the NP-Pep enriched cTnI over threefold ($p < 0.0001$) relative to the loading mixture (Fig. 2d and Supplementary Fig. 14b, c). High-resolution top-down MS analysis of the intact cTnI proteoforms demonstrated the high reproducibility of this process (Fig. 2e and Supplementary Figs. 14d and 15) and directly revealed all endogenous cTnI proteoforms present in the donor heart sample: bisphosphorylated cTnI with C-terminal truncation (ppcTnI[1–207]), unphosphorylated cTnI, monophosphorylated cTnI (pcTnI), and bisphosphorylated cTnI (ppcTnI) (Supplementary Table 5).

To illustrate the unique advantages of the NP-Pep system for cTnI enrichment over the existing immunoaffinity purification systems, we next compared the cTnI enrichment performance of the NPs against a conventional agarose-based solid support functionalized with the same high affinity cTnI-binding peptide (Agarose-Pep) and an anti-cTnI monoclonal antibody (M46; Agarose-mAb) possessing a similar cTnI-binding epitope (amino acids 130–145) to the high affinity peptide (Fig. 2f–h and Supplementary Fig. 17, details in the "Methods"). This cTnI enrichment comparison corrects for the bias due to affinity ligand and allows direct assessment of cTnI enrichment performance as a result of the choice of materials system. We evaluated the cTnI enrichment performance of the NP and Agarose platforms from sarcomeric extracts containing low amounts of cTnI (<700 ng) obtained from three different human heart samples: a donor heart with normal cardiac function (donor), a heart with dilated cardiomyopathy (diseased), and a postmortem heart with no known history of cardiac dysfunction (postmortem). SDS-PAGE analysis revealed an intense cTnI band present in the NP-Pep elution mixtures (upper panels of Fig. 2f–h), visually

demonstrating effective cTnI enrichment by the NP-Pep across each of the three human heart samples. Comparatively, the Agarose-Pep and Agarose-mAb showed reduced cTnI enrichment in the same SDS-PAGE (upper panels of Fig. 2f–h and Supplementary Fig. 17). These results highlight the unique advantages that these functionalized NPs possess for protein enrichment over antibody-based platforms: (a) comparable size and diffusion kinetics to proteins, promoting effective capture of low-abundance proteins within complex mixtures[20,47,48]; (b) high surface area per volume and high number of flexible binding sites, contributing to high ligand density and binding efficacy[49,50]; (c) inexpensive, simple, fast, and scalable synthesis with minimal batch-to-batch variability[35], in contrast to antibodies[14,15,51–53].

Furthermore, top-down LC/MS coupling reversed-phase liquid chromatography (RPLC) to high-resolution MS (see "Methods") revealed the molecular details of the endogenous cTnI proteoforms found in the loading mixture (L), flow through (F), and elution (E) for each heart sample (lower panels of Fig. 2f–h). We were able to unambiguously identify and characterize the various cTnI proteoforms inherent to each of the heart samples by accurate mass measurements (lower panels of Fig. 2f–h and Supplementary Table 6). We found ppcTnI to be the predominant cTnI proteoform in the non-diseased donor heart (Fig. 2f), whereas non-phosphorylated cTnI was found to be the most abundant cTnI proteoform in the dilated cardiomyopathy heart (Fig. 2g), consistent with previous reports suggesting phosphorylation of cTnI as a potential biomarker for heart diseases[27,31,32]. We also identified and characterized C-terminal truncated cTnI proteoforms found in the postmortem samples (Fig. 2h), which is consistent with our previous report[32]. Despite the rich diversity of cTnI proteoforms between the various heart samples, the NP-Pep was capable of faithfully retaining the endogenous cTnI proteoform distribution initially present in the donor (Fig. 2f), dilated cardiomyopathy (Fig. 2g), and postmortem (Fig. 2h) heart samples both before and after enrichment (Supplementary Fig. 18). Collectively, these results suggest that NP-Pep can be used as an effective antibody replacement for cTnI enrichment.

**Enrichment of low-abundance proteoforms from human serum.** Critically, MS-based detection of low-abundance cTnI proteoforms directly in the blood remains an unsolved challenge[27] due to the exceedingly complex and large dynamic range (~$10^{12}$) of the blood proteome[10] combined with the presence of highly abundant blood proteins such as HSA (~60% of total blood mass and >10 orders of magnitude in abundance than circulating cTnI)[9,10]. To this end, we performed cTnI enrichment from human serum spiked in with the cTnI extracted from the same donor, dilated cardiomyopathy, and postmortem heart samples (*vide supra*). The endogenous cTnI obtained from different human heart samples were used as the reference cTnI sources and served to mimic circulating cTnI found in the blood. SDS-PAGE analysis revealed the striking contrast in cTnI enrichment performance and nonspecific blood protein resistance between the NP-Pep versus the conventional agarose platform (Fig. 3a). Importantly, the NP-Pep demonstrated impressive resistance to nonspecific adsorption from the highly abundant HSA in human serum, in contrast to the Agarose-Pep and Agarose-mAb. Although cTnI was undetectable in the original serum mixture due to its low abundance compared to the higher-abundance blood proteins, the NP-Pep elution fraction showed significant cTnI enrichment compared to the original serum mixture (Fig. 3a) and nearly complete depletion of high-abundance proteins such as HSA (Supplementary Fig. 19). SDS-PAGE analysis of cTnI enrichment from human serum spiked with dilated cardiomyopathy and postmortem cTnI-containing sarcomeric extracts also revealed consistent HSA depletion by the NP-Pep, in contrast with the Agarose-Pep and

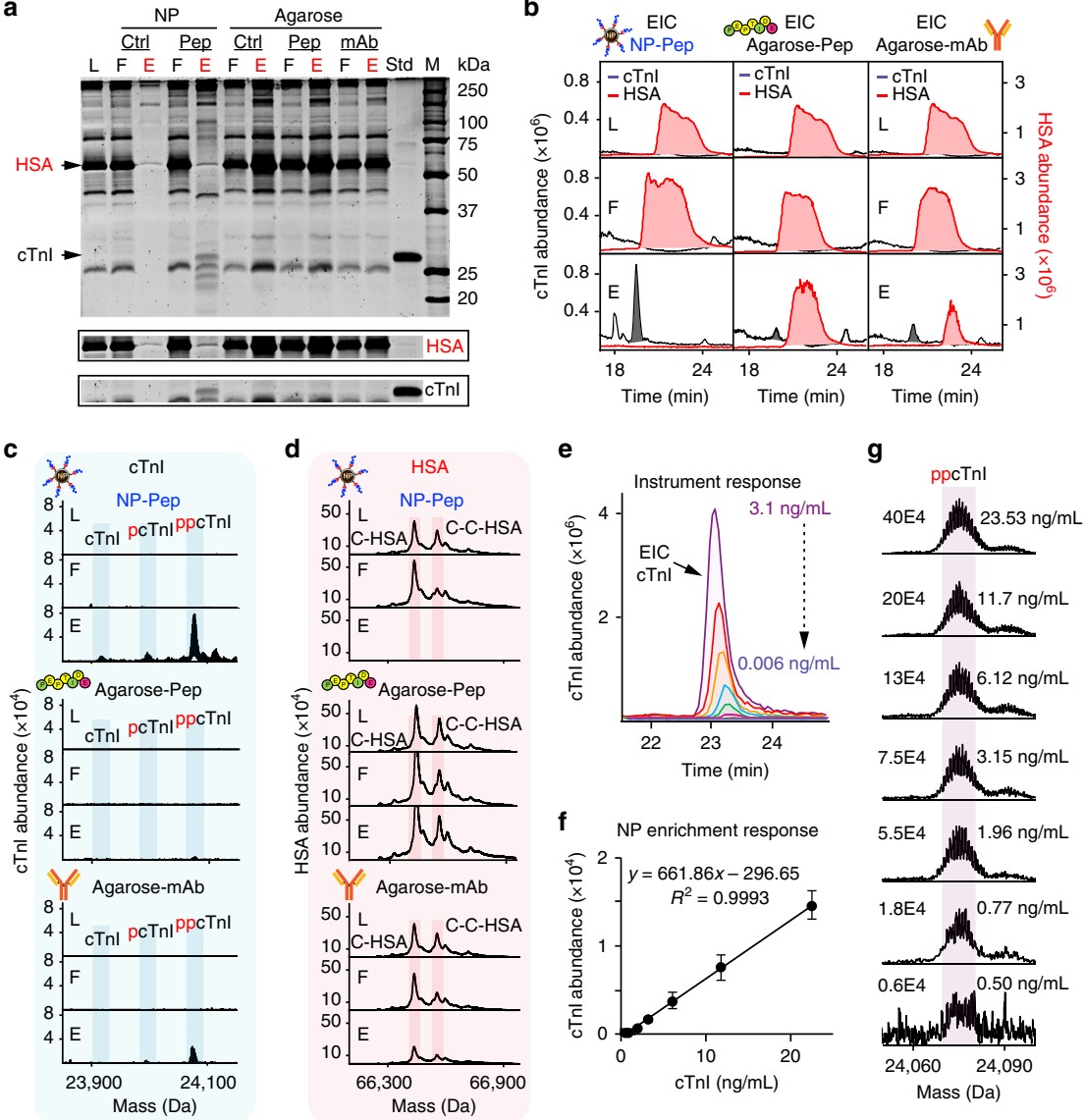

**Fig. 3 Direct enrichment of cTnI from human serum by NP-Pep with high specificity and sensitivity.** Human serum with spike-in cTnI was used for the evaluation of cTnI enrichment performance using NP-Pep and the conventional agarose-mAb. **a** SDS-PAGE visualizing the effectiveness of cTnI enrichment from human serum using different affinity platforms. The cTnI (306 ng) was obtained from a sarcomeric protein extract from a healthy donor heart and spiked into human serum (10 mg). Loading mixture (L), flow through (F), and elution mixture after enrichment (E) were equally loaded on the gel (500 ng) using NP-Ctrl, NP-Pep, Agarose-Ctrl, Agarose-Pep, and Agarose-mAb. The NP-Pep enabled nearly complete depletion of human serum albumin (HSA), allowing for more specific enrichment of cTnI from human serum. **b** Normalized extracted ion chromatograms (EICs) of cTnI (black) and HSA (red) corresponding to the NP-Pep, Agarose-Pep, and Agarose-mAb shown in (**a**). Normalized deconvoluted mass spectra corresponding to enriched cTnI (**c**) and depleted HSA (**d**), illustrating the abundance of cTnI and HSA before and after enrichment using NP-Pep, Agarose-Pep, and Agarose-mAb corresponding to (**a**, **b**). **e** Normalized EICs illustrating the MS response for cTnI (3.1–0.006 ng/ml) obtained from a human heart tissue extract ($n = 2$ independent experiments) using a Bruker maXis II ETD. **f** Evaluation of the sensitivity performance of the nanoproteomics assay. MS response against cTnI concentration is shown for various NP-Pep enrichment fractions from serum mixtures spiked with cTnI (22.53–0.50 ng/mL). The deconvoluted peak intensities of ppcTnI were used for the analyses. Data are representative of $n = 3$ independent experiments with error bars indicating standard error of the mean. **g** Representative deconvoluted mass spectra illustrating the MS response for ppcTnI corresponding to the plot in (**f**). M. protein marker, Std. endogenous cTnI protein standard, p phosphorylation, pp bisphosphorylation, HSA human serum albumin, C-HSA cysteinylated human serum albumin, C-C-HSA doubly cysteinylated human serum albumin, NP-Ctrl unfunctionalized NP, NP-Pep high affinity peptide-functionalized NP, Agarose-Ctrl unfunctionalized agarose, Agarose-Pep high affinity peptide-functionalized agarose, Agarose-mAb antibody (mAb M46) functionalized with agarose. Source data are provided as a Source Data file.

Agarose-mAb (Supplementary Fig. 20). This is further demonstrated by analysis of extracted ion chromatograms (EICs) corresponding to HSA and cTnI from the same serum samples in both the NP-Pep and the agarose platforms (Fig. 3b and Supplementary Figs. 21a and 22a).

Unlike the NP-Pep, the agarose platforms retained a large amount of HSA in the elution fraction, which likely limited its enrichment efficacy for cTnI. Importantly, the deconvoluted top-down mass spectra of the serum spike-in cTnI samples illustrate not only did the NP-Pep enrich bisphosphorylated cTnI

(ppcTnI), but also it was able to capture unphosphorylated cTnI and monophosphorylated cTnI (pcTnI) present in lower abundance from the non-diseased donor heart (Fig. 3c top panel). In contrast, minimal cTnI proteoforms were detected after enrichment by Agarose-Pep and Agarose-mAb, respectively (Fig. 3c bottom panels and Supplementary Figs. 21b and 22b). Meanwhile, the deconvoluted top-down mass spectra for HSA (Fig. 3d and Supplementary Figs. 21c and 22c) further confirmed the significant depletion of HSA by the NP-Pep compared to the Agarose-Pep and Agarose-mAb, where HSA remains in high abundance. These serum cTnI enrichment results illustrate the major benefits of functionalized NPs compared to current antibody-based approaches: (1) NPs are capable of reproducibly and effectively enriching cTnI (Supplementary Fig. 23) across all tested serum samples without the use of antibody, thus overcoming the intrinsic antibody-related limitations (*vide supra*)[14,15,51–53]. (2) NPs retain all endogenous cTnI proteoforms from serum without introducing artifactual modification. (3) NPs show highly reproducible and effective depletion of HSA concurrently with cTnI enrichment. Although the NP-Pep demonstrates highly effective HSA depletion, there are some nonspecific proteins still retained from the serum enrichment (Fig. 3a and Supplementary Fig. 20). To investigate these coeluted proteins, we performed a detailed analysis of all top-down LC/MS proteins identified in the NP-Pep serum elution mixtures (Supplementary Fig. 24). cTnI was demonstrated to be confidently identified and consistently captured in all NP-Pep serum enrichment trials (Supplementary Fig. 24).

Following the determination of the absolute cTnI concentration using an ELISA (Supplementary Fig. 25), we calculated the limit of detection (LOD) for cTnI using top-down MS by injecting controlled amounts of purified cTnI (3.1–0.006 ng) from the donor heart sample used in Figs. 2 and 3a–d. Top-down RPLC/MS with a CaptiveSpray (CS) ionization source fitted to a maXis II ETD mass spectrometer was sufficiently sensitive to detect cTnI with a LOD (3.3 σ/s) as low as 0.06 ng/mL (Fig. 3e and Supplementary Fig. 26). After demonstrating the capability of top-down MS for accurate detection and analysis of cTnI at low concentrations, we next evaluated the detection sensitivity of the current NP-Pep platform for cTnI enrichment from human serum. Notably, this integrated nanoproteomics method with specific enrichment of cTnI using the NP-Pep followed by top-down MS analysis enabled the detection of cTnI proteoforms from human serum at with a low LOD of 0.75 ng/mL (Fig. 3f, g and Supplementary Fig. 27). Note that the estimated levels of cTnI released after cardiac injury events such as AMI are typically in the range of 0.5–50 ng/mL[25,27]. Hence, this nanoproteomics approach enables analysis of cTnI proteoforms from human serum at clinically relevant concentrations. In addition, we evaluated the serum cTnI enrichment performance of the NP-Pep compared to the Agarose-mAb by ELISA detection of the cTnI amount before and after enrichment (Supplementary Fig. 28). The NP-Pep demonstrated a high cTnI enrichment factor (115-fold) and the serum cTnI percent recovery of the NP-Pep is ~51% (threefold higher than the Agarose-mAb, ~17%). We believe that the cTnI percent recovery can be further improved with future optimizations in automating the NP-Pep enrichment workflow to reduce sample handling and transfer steps which may result in unnecessary protein loss. Furthermore, additional instrumentation improvements in top-down MS[8] will further improve the LOD of the platform toward the diagnostic cutoff value used by contemporary cTnI ELISA (≤0.04 ng/mL).

**Proteoform-resolved analysis of cTnI from human serum.** Finally, we applied this nanoproteomics method to human serum

with a minimal amount of spike-in cTnI originating from various non-diseased donor, diseased, or postmortem heart samples (Supplementary Table 7) to mimic the diversity of circulating cTnI proteoforms present in patients' blood (Fig. 4). The effective cTnI enrichment by the NP-Pep allowed for the subsequent comprehensive analysis of cTnI proteoforms using top-down LC/MS (Fig. 4a) to reveal unique cTnI proteoform fingerprints in each human serum sample representing distinct pathophysiology (Fig. 4b and Supplementary Fig. 29). The deconvoluted mass spectra corresponding to cTnI proteoforms enriched from human serum samples revealed that the endogenous cTnI was primarily in its phosphorylated (mono- and bisphosphorylated) state in the two donor samples (Fig. 4b; panels i and ii), whereas drastic decrease of phosphorylated cTnI was detected in the two diseased samples with dilated cardiomyopathy (Fig. 4b; panels iii and iv). Moreover, cTnI degradation due to C-terminal proteolysis was clearly seen in the two postmortem samples (Fig. 4b; v and vi). Extensive modifications including phosphorylation and degradation of cTnI have previously been observed in serum samples from patients with AMI[28]. Tandem MS/MS analysis of the detected serum cTnI proteoforms was used to validate proteoform assignments across the various heart pathologies (Supplementary Fig. 30). Notably, cTnI phosphorylation is considered as an integral mechanism in cardiac homeostasis and has potential to reveal the functional status of the heart[54], whereas selective cTnI proteolytic degradation contributes to cardiac dysfunction in ischemia/reperfusion injury[29]. Altered PTM profiles of cTnI are associated with dysregulated cellular signaling during the onset and progression of diseases, thus disease-induced cTnI proteoforms are believed to have the potential to serve as the next-generation cardiac biomarkers for diagnosis of specific cardiovascular syndromes[27,32,33].

Here, we demonstrated that NP-Pep enables highly effective and unbiased cTnI proteoform enrichment across all tested serum samples with the faithful recapitulation of cTnI PTM profiles (Supplementary Fig. 31) despite the rich diversity of cTnI proteoforms present in each of these cardiac samples (Supplementary Table 8). The relative abundance of cTnI proteoforms present in the initial sarcomeric extracts did not significantly change after serum spike-in and incubation in pooled human serum followed by subsequent NP-Pep enrichment (Supplementary Fig. 31). The concurrent enrichment of cTnI and depletion of HSA together with the reliable preservation of the cTnI proteoform fingerprints during the NP enrichment solves the major challenge in the MS detection of cTnI proteoforms directly from the blood[27] and will enable the application of this top-down nanoproteomics strategy to clinical samples to eventually establish a proteoform-resolved cTnI assay.

## Discussion
Here we develop a proteoform-resolved method for the analysis of low-abundance proteins directly from serum enabled by top-down nanoproteomics. We demonstrate that carefully designed peptide-functionalized NPs can directly capture and enrich cTnI from human serum with high specificity and reproducibility, while simultaneously depleting highly abundant blood proteins, such as HSA. These NPs not only outperform conventional monoclonal antibody platforms for serum cTnI enrichment, but also faithfully and holistically preserve all endogenous cTnI proteoforms. Thus, these NPs can serve as replacements to conventional immuno-based techniques to overcome the antibody-related limitations in cTnI immunoassays and potentially address the current reproducibility crisis caused by antibodies in general[14–16,51,53]. This antibody-free approach can be leveraged in future clinical cTnI diagnostic assays. By further applying to a large human cohort,

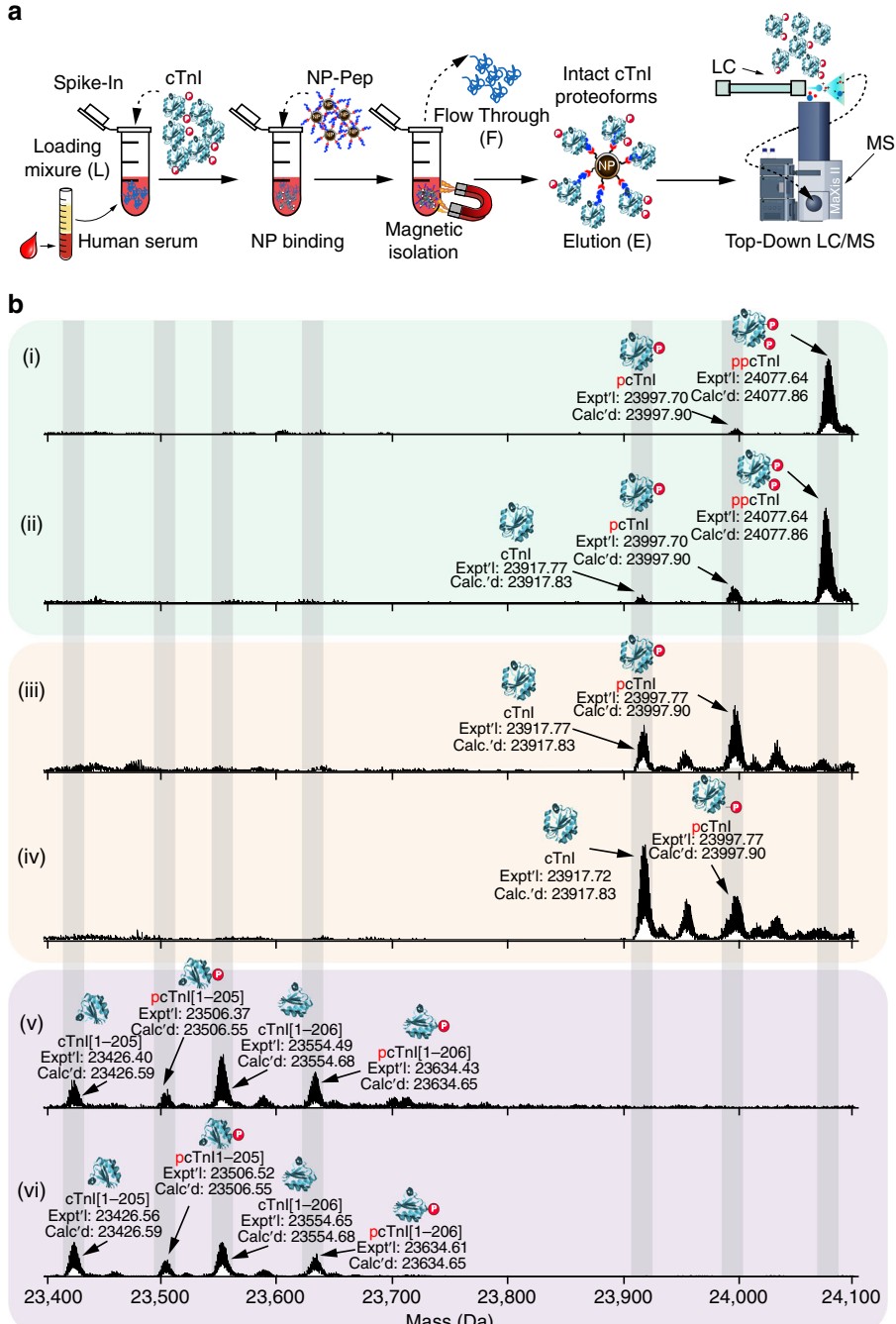

**Fig. 4 Nanoproteomics enables comprehensive analysis of cTnI proteoforms from human serum. a** Nanoproteomics assay utilizing NP-Pep for specific enrichment of cTnI from serum and subsequent top-down MS analysis of cTnI proteoforms. cTnI is first spiked into human serum to prepare the loading mixture (L). The NPs are then incubated with the serum loading mixture, the cTnI-bound NPs are magnetically isolated, the unwanted and nonspecific proteins are removed as flow through (F). The captured cTnI is then eluted and the final elution fraction after enrichment is analyzed by top-down LC/MS. **b** Deconvoluted MS corresponding to cTnI proteoforms enriched from human serum. The cTnI (~10–20 ng/mL) spiked in the human serum (10 mg) were extracted from various human hearts: (i) and (ii), donor hearts; (iii) and (iv), diseased hearts with dilated cardiomyopathy; (v) and (vi), postmortem hearts. cTnI proteoforms were identified using accurate intact mass measurement, using the most abundant mass based on the amino acid sequence of entry name TNNI3_human from the UniProtKB sequence database. p phosphorylation, pp bisphosphorylation. Data are representative of $n = 3$ independent experiments. A summary of the enriched proteoforms from serum and their respective mass measurements by top-down MS are listed in Supplementary Table 8. Source data are provided as a Source Data file.

patient blood samples can be analyzed to comprehensively detect all cTnI proteoforms and establish the relationships between cTnI proteoforms and underlying disease etiology[27]. Ultimately, this nanoproteomics strategy could enable next-generation precision medicine approaches for comprehensive cTnI analysis toward accurate diagnosis, better risk stratification, and improved outcome assessment of patients presenting with various cardiovascular syndromes.

Beyond cTnI, we expect that this scalable and reproducible top-down nanoproteomics approach can be generally applied to other

low-abundance plasma/serum proteins of interest, such as cytokines or growth factors, if the NPs can be functionalized with a suitable affinity reagent[55,56]. With the exciting recent advances in phage display libraries[57], in silico techniques[37], and designing aptamers/affimers largely owing to advancements in high-throughput methods for systemic evolution of ligands by exponential enrichment[58,59], the design and selection of effective affinity reagents for targeted protein analysis is more accessible than ever before. This nanoproteomics strategy is capable of providing previously unachievable molecular details of low-abundance serum proteins in general and can serve as an enabling technology to comprehensively map the proteoform landscape.

## Methods

**Materials and reagents**. All chemicals and reagents were purchased from MilliporeSigma (St. Louis, MO, USA) and used as received without further purification unless otherwise noted. Sodium oleate (97%) was purchased from Tokyo Chemical Industry America (Portland, OR, USA). The high affinity cTnI peptide (95%) with C-terminal Cys residue (HWQIAYNEHQWQC) and the negative peptide (95%) with C-terminal Cys residue (HWNMAANEHMQWC) were purchased from GenScript USA Inc (Pescataway, NJ, USA). (3-aminopropyl)triethoxysilane (APTES) was purchased from Gelest (Morrisville, PA, USA). Human male AB serum (H4522) was purchased from MilliporeSigma. Human cTnI-C monoclonal antibody (M46, cat. # sc-52277) and phosphatase inhibitor cocktail A (cat.# sc-45044) were purchased from Santa Cruz Biotechnology, Inc (Dallas, TX, USA). N-hydroxysuccinimide activated agarose slurry (cat.# 26200) and Halt[TM] Protease inhibitor cocktail were purchased from ThermoFisher Scientific (Rockford, IL, USA). Human cTnI ELISA kit (AccuBind® ELISA, cat. # 3825-300) was purchased from Monobind Inc (Lake Forest, CA, USA). Extraction solutions were made in nanopure deionized water (H₂O) from Milli-Q® water (MilliporeSigma). Bradford protein assay reagent was purchased from Bio-Rad (Hercules, CA, USA). In all, 12.5% gel (10 or 15 comb well, 10.0 × 10.0 cm, 1.0 mm thick) for SDS-PAGE was homemade. DynaMag[TM]-2 Magnet and Thermo Scientific[TM] Cimarec+[TM] stirring hotplate were purchased from ThermoFisher Scientific. Amicon, 0.5 mL cellulose centrifugal filters with a molecular weight cutoff (MWCO) of 10 kDa were purchased from MilliporeSigma.

**Synthesis of 3-butynoic acid**. 3-Butynoic acid was synthesized following a previously reported procedure[36], with modification. Deionized water (135 mL) was added to a 500 mL single neck round bottom flask fitted with magnetic stirrer bar. Seventy percent HNO₃ (0.170 mL, 0.05 eq., 2.5 mmol), Na₂Cr₂O₇ · 2H₂O (0.15 g, 0.01 eq., 0.5 mmol), and NaIO₄ (34.2 g, 3.2 eq., 160 mmol) were subsequently added to the flask and the mixture was stirred vigorously on an ice bath for 15 min. 3-Butyn-1-ol (3.50 g, 1 eq., 50 mmol) was diluted with chilled deionized water (135 mL), added to the flask slowly, and stirred overnight. After this time, the reaction mixture was filtered to remove insoluble salts, and the product was extracted from the aqueous phase using diethyl ether (80 mL × 5). The organic fractions were combined, dried over anhydrous magnesium sulfate, and concentrated under rotary evaporation to give an orange viscous liquid. Subsequent washing with pentane (200 mL × 3) and further concentration under rotary evaporation times yielded 3-butynoic acid as an off-white solid (2.19 g, 52% yield). ¹H NMR (500 MHz, Chloroform-d) δ (ppm) 3.38 (d, 2H, J = 2.7 Hz), 2.25 (t, 1H, J = 2.7 Hz); ¹³C NMR (125 MHz, Chloroform-d) δ (ppm) 173.8, 74.8, 72.4, 25.6.

**Synthesis of N-(3-(triethoxysilyl)propyl)buta-2,3-dienamide**. 3-Butynoic acid (2.10 g, 1 eq., 25 mmol), 2-chloro-1-methylpyridinium iodide (6.39 g, 1 eq., 25 mmol), and dichloromethane (250 mL) were added to an oven dried 500 mL three-neck flask fitted with a reflux condenser. The solution was heated to reflux under N₂. In a separate flask, APTES (5.53 g, 1 eq., 25 mmol) and N,N-diisopropylethylamine (6.46 g, 2 eq., 50 mmol) were diluted with dichloromethane (125 mL), and added to the refluxing three-neck flask by a syringe. The reaction mixture was allowed to reflux for 1 h, and then N,N-diisopropylethylamine (6.46 mL, 2 eq., 50 mmol) was additionally added to effectively isomerize the propargylic isomer to N-(3-(triethoxysilyl)propyl)buta-2,3-dienamide. The reaction was allowed to reflux overnight, and the product was concentrated by rotary evaporation. The crude product was redispersed in ethyl acetate, centrifuged at 5500 × g for 5 min, and the supernatant was concentrated by rotary evaporation. The product was purified with flash column chromatography using a gradient of 50:50 ethyl acetate: n-hexane to 100:0 ethyl acetate: n-hexane to yield BAPTES as a clear, orange oil (4.80 g, 67% yield). R_f (Ethyl Acetate) = 0.67; ¹H NMR (500 MHz, Chloroform-d) δ (ppm) 5.99 (s, 1H), 5.62 (t, J = 6.6 Hz, 1H), 5.20 (d, J = 6.7 Hz, 2H), 3.83 (q, J = 7.0 Hz, 6H), 3.31 (q, J = 6.7 Hz, 2H), 1.66 (m, 2H), 1.24 (t, J = 7.0 Hz, 9H), 0.66 (m, 2H); ¹³C NMR (125 MHz, Chloroform-d) δ (ppm) 211.56, 164.33, 91.02, 80.33, 58.47, 42.02, 23.02, 18.30, 7.66; ESI-MS for C₁₃H₂₅NO₄Si [M + H]⁺ observed: 288.162 m/z, [M + H]⁺ calculated: 288.162 m/z.

**Synthesis of iron-oleate precursor**. Iron oleate was synthesized using a previously established method[34]. In a typical synthesis, iron (III) chloride hexahydrate (10.8 g, 40 mmol) was first dissolved in a mixture of 80 mL ethanol and 60 mL nanopure water in a three-neck round bottom flask (500 mL) containing a Teflon-coated egg-shaped (1 − 1/4″ × 5/8″) magnetic stir bar. Sodium oleate (36.5 g, 120 mmol) was then quickly added to the iron chloride solution along with 140 mL n-hexane. The resulting solution was then allowed to stir until the sodium oleate was completely dissolved. Afterwards, the reaction solution was heated to 70 °C for a 4-h reflux under a N₂ blanket. Upon completion, the reaction solution was cooled to room temperature, and the upper organic layer containing the iron oleate was washed three times with 30 mL nanopure water in a 250 mL separatory funnel. After washing, the iron oleate was concentrated by rotary evaporation. Finally, the resulting iron oleate was transferred into a 100 mL round bottom flask, connected to a Schlenk line, and placed under vacuum overnight. For storage, the iron oleate was well sealed in a glass vial and placed in a desiccator.

**Synthesis of 8 nm magnetite (Fe₃O₄) nanocrystals**. Iron oleate (10 mmol, 9.0 g) and oleic acid (5.5 mmol, 1.56 g) were added into a three-neck round bottom flask (250 mL) with a solvent mixture of 1-octadecene:1-tetradecene (40:10 g). The mixture was stirred and degassed on a Schlenk line at 110 °C for 3 h. The mixture was then placed under nitrogen flow and the reaction solution was heated to 300 °C at a heating rate of 3.3 °C/min using a temperature controller. Reaction time was counted starting from when 300 °C was reached. After 1 h, the reaction solution was quickly cooled to room temperature by blowing air across the reaction flask. The resulting iron-oxide nanocrystals were precipitated using ethanol, isolated via centrifugation (5500 × g, 20 min), and then washed with three precipitation/redispersion cycles (5500 × g, 20 min) using ethanol. The resulting nanocrystals were then dried under vacuum, weighed, and redispersed in n-hexane at a concentration of 20 mg/mL for further use.

**Synthesis of Fe₃O₄-BAPTES NPs (NP-BAPTES)**. In a typical optimized large-scale synthesis of silane functionalized NPs, Fe₃O₄ NPs (6 mL from a 20 mg/mL stock) were added to anhydrous n-hexane (300 mL) in a 500 mL round bottom flask equipped with a Teflon-coated egg-shaped magnetic stir bar (1 − 1/4″ × 5″) to achieve a total NP concentration of 0.4 mg/mL. After the reaction mixture was heated to 60 °C with stirring (900 rpm), BAPTES (1.65 mL) was added dropwise to the flask for a 0.55% (v/v) total concentration of trialkoxysilane reagent, followed by dropwise addition of a small amount of acetic acid (30 μL) for an acidic catalyst concentration of 0.01% (v/v). After reaction under stirring for 24 h, the precipitate was collected and washed one time with n-hexane, one time with n-hexane/acetonitrile (v/v, 4:1), and one more time with n-hexane via centrifugation (5500 × g, 10 min) to remove excess silane molecules and surfactants. The NPs were then dried under vacuum for later use.

**Synthesis of Fe₃O₄-BAPTES-Peptide NPs (NP-Pep)**. 10 mg of NP-BAPTES was added to a 4-dram vial and dispersed in 2 mL of acetonitrile. 10 mg of cTnI-binding peptide (HWQIAYNEHQWQC) was added to a separate 4-dram vial and dissolved in 8 mL of nanopure water. The pH of the peptide solution was adjusted to pH 8.0 by the addition of 75 μL of 1.0 M ammonium carbonate buffer pH 9.0 during simultaneous water bath sonication. The pH-adjusted peptide solution was added into the 4-dram vial containing the NP dispersion under water bath sonication. The NP reaction mixture was allowed to react under sonication for 1 h and later collected into Eppendorf tubes for washing. The peptide-functionalized NPs were washed three times with nanopure water via centrifugation (15,000 × g, 5 min) and subsequently isolated magnetically with a DynaMag to remove unreacted peptide. The resulting peptide-functionalized NPs were redispersed in nanopure water at a concentration of 5 mg/mL.

**Material characterization**. TEM samples were prepared by pipetting a 10 μL drop of as-synthesized NPs at a concentration of 0.125 mg/mL onto a copper TEM grid with lacey carbon film. TEM was conducted on a FEI T12 microscope (FEI Company, Hillsboro, OR, USA) operated at 120 kV, equipped with a Gatan CCD image system with digital micrograph software program. Transmission FTIR spectroscopy measurements were recorded on a Bruker Equinox 55 FTIR spectrometer (Bruker Optik GmbH, Ettlingen, Germany) in the range of 4000–400 cm⁻¹ at 2 cm⁻¹ resolution on NPs in a potassium bromide (KBr) pellet, at a sample mass loading of 0.33 wt.%. TGA was carried out using a TA Instruments Q500 (TA Instruments, New Castle, DE, USA) thermal analysis system under a N₂ atmosphere and at a constant heating rate of 10 °C/min from 100 to 600 °C. All samples were first heated to 100 °C and held at that temperature for 3 min to remove adsorbed water. ζ-potential measurements was carried out using a Malvern Zetasizer Nano.

**Small molecule analysis by FTICR-MS**. Small molecule samples (<3 kDa) were diluted 500-fold in 50:50:0.1 (acetonitrile:water:formic acid) MS-grade solvent for positive electrospray ionization mode analysis. Samples were analyzed by direct infusion using a TriVersa Nanomate system (Advion BioSciences, Ithaca, NY, USA) coupled to a solariX XR 12-Tesla Fourier Transform Ion Cyclotron Resonance mass spectrometer (FTICR-MS, Bruker Daltonics, Bremen, Germany). For

the nano-electrospray ionization source using a TriVersa Nanomate, the desolvating gas pressure was set at 0.5 PSI and the voltage was set to 1.2–1.6 kV versus the inlet of the mass spectrometer. Mass spectra were acquired with an acquisition size of 1 M, in the mass range between 150 and 2000 $m/z$ (with a resolution of 270,000 at 400 $m/z$), and 50 scans were accumulated for each sample. Ions were accumulated in the collision cell for 0.05 s, and a time of flight of 0.500 ms was used prior to their transfer to the ICR cell. For collisionally activated dissociation tandem MS (MS/MS) experiments, the collision energy was varied from 10 to 20 V. Tandem mass spectra were output from the DataAnalysis software and analyzed using MASH software[60]. The methods described here correspond to the data presented in Supplementary Figs. 3, 7, and 8.

**Sarcomeric protein extraction from human cardiac tissue**. All protein extraction procedures were performed in a cold room (4 °C) using freshly prepared buffers. All human cardiac tissue was voluntarily obtained from the University of Wisconsin Hospital and Clinic with informed consent. Discarded myocardial tissues from patients who undergo cardiac surgery as well as donor heart tissue from the Organ Procurement Organization were used in this study. The procedure for the collection and de-identification of human cardiac tissue was approved by the Institutional Review Board of the University of Wisconsin—Madison. 500 mg of tissue was homogenized in wash buffer (5 mM NaH$_2$PO$_4$, 5 mM Na$_2$HPO$_4$, 5 mM MgCl$_2$, 0.5 mM EGTA, 0.1 M NaCl, 1% Triton X-100, 5 mM DTT, 1 mM PMSF, 1× HALT protease inhibitor cocktail, and 1× phosphatase inhibitor cocktail A, pH 7.4) using a Polytron electric homogenizer (Model PRO200; PRO Scientific, Oxford, CT, USA) on ice. The resulting homogenate was centrifuged at 10,000 × $g$ (Avanti J-25i; Beckman Coulter, Fullerton, CA, USA) for 10 min at 4 °C. After centrifugation, the supernatant was removed, and the pellet was washed once more with wash buffer. The wash supernatant was removed, and the pellet was resuspended in protein extraction buffer (0.7 M LiCl, 25 mM Tris, 5 mM EGTA, 0.1 mM CaCl$_2$, 5 mM DTT, 1 mM PMSF, 1× HALT protease inhibitor cocktail, and 1× phosphatase inhibitor cocktail A, pH 7.5), and the suspension was agitated on a nutating mixer (Thermo Scientific, Boston, MA, USA) at 4 °C for 45 min. The sarcomeric protein extract was centrifuged at 16,000 × $g$ (Centrifuge 5415R, Eppendorf, Hamburg, Germany) for 10 min at 4 °C and the resulting supernatant was again centrifuged at 21,000 × $g$ for 30 min to remove all tissue debris. The concentration of the tissue lysate was determined by Bradford protein assay. Samples were stored at −80 °C for later study. The buffers and reagent preparation for the methods described here correspond to the data presented in Supplementary Table 4.

**cTnI enrichment using NP-Pep from sarcomere extracts**. NP-Pep (5 mg) was redispersed in a 2 mL Eppendorf Protein Lo-Bind tube with equilibration buffer (50 mM Tris, pH 7.5, 150 mM LiCl). The NPs were then centrifuged at 15,000 × $g$ for 2 min at 4 °C, isolated from the solution using the DynaMag, and the supernatant was removed. Equilibration buffer was then added to the NPs, the mixture was sonicated and vortexed to prepare for protein loading. Protein loading mixture (L) from tissue extract was diluted to a final volume of 1 mL with a buffered solution (50 mM Tris, pH 7.4, 150 mM LiCl) to a total protein loading of 0.3 mg/mL and was added to the NP-Pep mixture, at an NP concentration of 5 mg/mL. After this mixture was agitated on a nutating mixer at 4 °C for 40 min, the NPs were centrifuged at 15,000 × $g$ for 2 min at 4 °C, and then isolated from the solution using the DynaMag. The supernatant was collected and saved as the flow-through (F) fraction. The isolated NPs were then washed three times with a wash buffer (50 mM Tris, pH 7.5, 300 mM NaCl; 0.20 mL/mg NP) following the same centrifugation and magnetic isolation steps to remove unbound, nonspecific proteins. To elute the bound cTnI, 500 μL of 200 mM glycine hydrochloride buffer (pH 2.2) was added. After centrifugation and magnetic isolation, the resulting supernatant was collected as the elution fraction (E). We found that the majority of proteins were eluted in the first elution fraction and that subsequent elutions using the same 200 mM glycine hydrochloride buffer or 1% SDS yielded minimal protein. Typically, only the first elution mixture was used for further analysis. All protein fractions (L, F, and E) were desalted prior to MS analysis using a 10 kDa MWCO filter (Amicon, 0.5 mL, cellulose, MilliporeSigma) and buffered exchanged using 0.2% formic acid in nanopure water. To evaluate the enrichment performance of the functionalized NPs, all collected fractions (L, F, and E) were equally loaded (500 ng), separated using a polyacrylamide gel (12.5%), and stained with SYPRO Ruby (ThermoFisher Scientific) fluorescent dye. The gel was imaged using a ChemiDoc™ MP Imaging System (170–8280; Bio-Rad, Hercules, CA, USA). Uncropped and unprocessed scans of all blots and gels are available in the Source Data file.

**cTnI enrichment using agarose beads from sarcomere extracts**. mAb or peptide was conjugated to the agarose beads following the manufacturer's recommendations and blocked with a 1.0 M ethanolamine solution, pH 7.4. Protein loading mixture (L) was incubated with 500 μL of mAb-conjugated or peptide-conjugated agarose beads in a disposable affinity column for 40 min on a nutating mixer at 4 °C. After incubation, the supernatant was collected and saved as the flow-through (F) fraction. The agarose beads were then washed three times with 1 mL of wash buffer (50 mM Tris pH 7.4, 300 mM NaCl) to elute unbound proteins. Subsequently, the bound proteins were eluted using four equal fractions of 500 μL of 200 mM glycine hydrochloride (pH 2.2). The four elution fractions (E)

were pooled and all protein fractions (L, F, and E) were desalted prior to MS analysis using a 10 kDa MWCO filter (Amicon, 0.5 mL, cellulose, MilliporeSigma) and buffered exchanged using 0.2% formic acid in nanopure water.

**cTnI enrichment using the NP-Pep from human serum**. Serum loading mixtures were prepared by serially spiking in human sarcomeric protein extracts into 200 μL of serum (10 mg protein) from human male AB plasma and diluting the serum mixture to a final volume of 1 mL using a buffered solution (50 mM Tris pH 7.4, 1.0 M LiCl). The concentrations of cTnI spiked into serum samples were measured using a cTnI AccuBind ELISA kit (Monobind Inc). The cTnI spiked human serum loading mixture (L) was incubated with NP-Pep (5 mg) in a 2 mL Eppendorf Lo-Bind tube. Typically, a relative loading of 0.5 mg NP-Pep per 1 mg of human serum was determined to be sufficient for effective serum cTnI enrichment. After this mixture was agitated on a nutating mixer at 4 °C for 40 min, the NPs were centrifuged at 15,000 × $g$ for 2 min at 4 °C, and then isolated from the solution using a DynaMag. The supernatant was collected and saved as the flow-through (F) fraction. The NPs were then washed three times with 1 mL of wash buffer (50 mM Tris pH 7.4, 700 mM NaCl) following the same centrifugation and magnetic isolation steps to remove unbound, nonspecific proteins. To elute the bound cTnI, 500 μL of 200 mM glycine hydrochloride buffer (pH 2.2) was added. We found that the majority of proteins were eluted in the first elution fraction and that subsequent elutions using the same 200 mM glycine hydrochloride buffer or 1% SDS yielded minimal protein. Typically, only the first elution mixture was used for further analysis. After centrifugation and magnetic isolation, the resulting supernatant was collected as the elution fraction (E). All protein fractions (L, F, and E) were desalted prior to MS analysis using a 10 kDa MWCO filter (Amicon, 0.5 mL, cellulose, MilliporeSigma) and buffer exchanged using 0.2% formic acid in nanopure water.

**cTnI enrichment using agarose beads from human serum**. Serum loading mixtures were prepared by serially spiking in human sarcomeric protein extracts into 200 μL of serum (10 mg protein) from human male AB plasma and diluting the serum mixture to a final volume of 1 mL using a buffered solution (50 mM Tris pH 7.4, 1.0 M LiCl). mAb or peptide was conjugated to the agarose beads following the manufacturer's recommendations and blocked with a 1.0 M ethanolamine solution, pH 7.4. The serum loading mixture (L) was added to 500 μL of mAb-conjugated or peptide-conjugated agarose beads in a disposable affinity column and agitated for 40 min on a nutating mixer at 4 °C. After incubation, the supernatant was collected and saved as the flow-through (F) fraction. The agarose beads were then washed three times with 1 mL of wash buffer (50 mM Tris pH 7.4, 700 mM NaCl) to elute unbound proteins. Subsequently, the bound proteins were eluted using four equal fractions of 500 μL of 200 mM glycine hydrochloride (pH 2.2). The four elution fractions (E) were pooled and all protein fractions (L, F, and E) were desalted prior to MS analysis using a 10 kDa MWCO filter (Amicon, 0.5 mL, cellulose, MilliporeSigma) and buffered exchanged using 0.2% formic acid in nanopure water.

**Comparison of cTnI enrichment from sarcomere extracts**. The enrichment workflow was performed in a cold room held at 4 °C to minimize possible artifactual protein modifications, such as oxidation. NP-Control (NP-BAPTES), NP-Pep (Fe$_3$O$_4$-BAPTES-Peptide), Agarose-Control beads (no coupling), Agarose-Pep beads, and Agarose-mAb beads were incubated with sarcomeric extract obtained from completely de-identified healthy donor heart tissue, diseased heart tissue of dilated cardiomyopathy, and postmortem heart tissue containing 306, 482, and 465 ng cTnI in the protein extract, respectively. cTnI values were determined by ELISA quantification. The loading mixture (L), flow through (F), and elution mixture (E) were collected, desalted using a 10 kDa MWCO filter, and buffer exchanged prior to SDS-PAGE gel analysis or liquid chromatography-mass spectrometry (LC-MS) analysis as described in the tissue enrichment protocol. The methods described here correspond to the data presented in Fig. 2f–k and Supplementary Figs. 17 and 18.

**Comparison of cTnI enrichment from serum spiked with cTnI**. The enrichment workflow was performed in a cold room held at 4 °C to minimize possible artifactual protein modifications, such as oxidation. NP-Control (NP-BAPTES), NP-Pep (Fe$_3$O$_4$-BAPTES-Peptide), Agarose-Control beads (no coupling), Agarose-Pep beads, and Agarose-mAb beads were incubated with in a protein loading mixture containing human male AB serum (10 mg) and sarcomeric tissue extract obtained from healthy donor heart tissue, diseased heart tissue of dilated cardiomyopathy, and postmortem heart tissue containing 306, 482, and 465 ng cTnI in the protein extract, respectively. cTnI values were determined by ELISA quantification. The loading mixture (L), flow through (F), and elution mixture (E) were collected, desalted using a 10 kDa MWCO, and buffer exchanged prior to SDS-PAGE gel analysis or LC-MS analysis as described in the tissue enrichment protocol. The methods described here correspond to the data presented in Fig. 2a–d and Supplementary Figs. 20–22.

**Typical reverse phase chromatography (RPC) procedure**. RPC was performed with a nanoACQUITY UPLC system (Waters Corporation; Milford, MA, USA). Mobile phase A contained 0.2% formic acid in nanopure water, and mobile phase B (MPB) contained 0.2% formic acid in 50:50 acetonitrile:isopropanol. Prior to

injection, protein samples were desalted by washing through 10 kDa MWCO filters using 0.2% formic acid in nanopure water six times. For each injection, 500 ng of desalted protein sample was loaded onto a BIOshell A400 C4, 3.4 μm, 15 cm × 200 μm capillary column (Supelco, Bellafonte, PA, USA). The column was placed in a column heater set at 60 °C with a constant 3 μL/min flow rate. The RPC gradient consisted of the following concentrations of MPB: 10% MPB at 0 min, 10% at 5 min, 65% at 45 min, 90% at 50 min, held at 90% until 55 min, adjusted back to 10% at 55.1 min, and held at 10% until 60 min. Each run was 60 min long.

**Top-down MS analysis**. Samples eluted from RPC separation were ionized using a CS source (Bruker Daltonics) into a maXis II Q-TOF mass spectrometer (Bruker Daltonics) for online LC-MS and LC-MS/MS experiments. End plate offset and capillary voltage were set at 500 and 4000 V, respectively. The nebulizer was set to 0.3 bar, and the dry gas flow rate was 4.0 L/min at 200 °C. The quadruple low mass cutoff was set to 500 $m/z$ for MS and 200 $m/z$ for MS/MS. Mass range was set to 200–3000 $m/z$ and spectra were acquired at 1 Hz for LC-MS runs. All data were collected with otofControl 3.4 (Bruker Daltonics), analyzed and processed in DataAnalysis 4.3 (Bruker Daltonics). Maximum Entropy algorithm (Bruker Daltonics) was used to deconvolute all mass spectra with the resolution set to 80,000. Sophisticated Numerical Annotation Procedure peak-picking algorithm (quality factor: 0.4; signal-to-noise ratio (S/N): 3.0; intensity threshold: 500) was applied to determine the monoisotopic mass of all detected ions. Chromatograms in Fig. 3 and Supplementary Figs. 21 and 22 shown were smoothened by Gauss algorithm with a smoothing width of 1.67 s. To quantify protein expression across samples, the Top 5 most abundant charge states' ions (average ± 0.2 $m/z$) of all major proteoforms from the same protein were retrieved collectively as one extracted peak in the EIC[61]. The area under curve was manually determined for each protein isoform using DataAnalysis. To quantify protein modifications, the relative abundances of specific modifications were calculated as their corresponding percentages among all the detected protein forms in the deconvoluted averaged mass. Tandem mass spectra were output from the DataAnalysis software and analyzed using MASH software[60]. For the targeted collision-induced dissociation LC-MS/MS analysis shown in Supplementary Fig. 30, the collision energy was varied from 18 to 30 V and the quadruple low mass was set to 500 $m/z$ with a scan range of 200–3000 $m/z$. The total ion current corresponding to all obtained MS/MS signal was averaged across the LC retention window corresponding to cTnI, and the averaged MS/MS data were then directly imported into MASH Explorer for proteoform identification and sequence mapping. All the program-processed data were manually validated. Peak extraction was performed using a S/N ratio of 3 and a minimum fit of 60%, and all peaks were subjected to manual validation. Alignment-based MS-Align+ algorithm (TopPIC)[62] for intact protein identification based on protein spectrum matches was used to search against the UniprotSwissprot Human database, which was released on June 26, 2019. Fragment mass tolerance was set to 15 ppm. All identifications were validated with statistically significant $p$ and $E$ values (<0.01) and satisfactory numbers of assigned fragment (>10). A 15-ppm mass tolerance was used to match the experimental fragment ions to the calculated fragment ions based on amino acid sequence.

**Statistical analysis**. All statistical data were presented as the mean ± standard error of the mean (SEM) unless stated otherwise. Two-tailed Student's $t$ test was used for comparison of two groups to evaluate the statistical significance of variance for the validation of the simultaneous quantification of protein expression and modification changes. Differences among means were considered significant at $p < 0.05$. All error bars shown in the figures were based on SEMs unless stated otherwise.

**Reporting summary**. Further information on research design is available in the Nature Research Reporting Summary linked to this article.

## Data availability
All data generated or analyzed during this study are presented in this paper or in the Supplementary Information. All the raw data files or spectra are available upon request. In addition, proteomics data have been uploaded to PRIDE repository via ProteomeXchange with identifier PXD019712. All other data are available from the corresponding authors on reasonable request. Source data are provided with this paper.

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

## Acknowledgements

This research is supported by NIH R01 GM117058 (to S.J. and Y.G.). Y.G. would like to acknowledge NIH R01 GM125085, R01 HL096971, and S10 OD018475. T.N.T. also thanks the fellowship support from the NIH Chemistry-Biology Interface Training Program NIH T32GM008505. K.A.B. would like to acknowledge support from the Training Program in Translational Cardiovascular Science, T32 HL007936-19. We thank Prof. Timothy Kamp for critical reading of this paper and insightful discussions. We also thank Prof. Amish Raval and William Swain for the helpful discussions. Moreover, we would like to thank Dr. Takushi Kohmoto, Dr. Ken Young, Elizabeth Bayne, James Anderson, and Carrie Sparks for helping collect the human heart samples and providing the clinical information.

## Author contributions

T.N.T. and D.S.R. contributed equally to this work. Y.G. and S.J. conceived the study and supervised the research; T.N.T., D.S.R., K.A.B., Y.Z., T.M.G.-A., S.J., and Y.G. designed the research; T.N.T. and D.S.R. performed the research; T.N.T., D.S.R., K.A.B., Y.Z., B.C., Z.W., and S.D.M. analyzed the data; T.N.T., D.S.R., S.J., and Y.G. wrote the paper with input from all co-authors.

## Competing interests

The University of Wisconsin—Madison has filed a provisional patent application serial No. 62/949,869 (December 18, 2019) on the basis of this work. Y.G., S.J., T.N.T., and D.S.R. are named as the inventors on the provisional patent application. The other authors declare no competing interests.

## Additional information

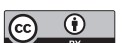

