## [Peer Review File · Nature Communications]

REVIEWER COMMENTS

Reviewer #1 (Remarks to the Author):

Top-down mass spectrometry (MS) proteomics plays a critical role in identifying post-translational modification of proteins. However, the detection and analysis of low-abundance proteins in plasma/serum by MS has proven challenging. The authors developed a nanoproteomics platform by conjugating superparamagnetic nanoparticles with targeting peptides to enrich cardiac troponin I (cTnI), a biomarker for cardiovascular disease, in serum samples. The nanoproteomics platform showed excellent cTnI enrichment relative to other abundant proteins in plasma. The authors performed a thorough demonstration of the utility of this platform, showing diverse cTnI proteoforms and establishing the proteoform-pathophysiology relationship. This is compelling work that could be further improved by addressing the following critiques before publication in Nature Communications.

1. From Fig 1c-1e, it appears that there may be Fe₃O₄-peptide aggregation. Therefore, the reviewer suggests adding more experiments on the dispersibility and size uniformity of Fe₃O₄-peptide, as this may influence surface area and cTnI enrichment.
2. While SDS-PAGE provides a gross measure of assay reproducibility, it has important limitations. It would be more compelling to see bottom-up MS data of three replicates (NP-Targeting Peptide) from the serum with data presentation in a Venn diagram to show reproducibility.
3. Fig. 11 demonstrates the authors' claim that the salt (NaCl) concentration of the wash buffer was a critical tunable parameter to promote effective cTnI enrichment. It would be helpful to test that hypothesis and explain this phenomenon with additional relevant experiments. In addition, the authors should also determine the zeta potential of the NP-Pep, since the electric charge intensity could influence cTnI enrichment when using different NaCl concentrations as the washing buffer.
4. When the authors sought to evaluate the cTnI enrichment performance of NP-Pep and measure cTnI enrichment, they first used ELISA to briefly determine the cTnI ratio. Does this mean that the authors should also use ELISA to determine the ratio and then use the corresponding concentration of NP-Pep for cTnI enrichment? What determines the concentration of NP-Pep used?
5. Although the targeting peptide can specifically adsorb cTnI proteoforms with depletion of human serum albumin, there are still other nonspecific proteins on the SDS-PAGE gel images that are normal. The authors should discuss this in more depth. In addition, could the authors analyze the most abundant proteins (e.g., the top 10) and their accumulation percentages? Does the size of nonspecific proteins and the size difference between NPs and nonspecific proteins affect enrichment performance?
6. The targeting peptide, which has an excellent binding affinity (0.27 nM), lends the iron oxide nanoparticles outstanding capacity to enrich cTnI proteoforms. In human blood samples, the authors

use the spike-in cTnI samples to test their platform. With known spike-in cTnI samples, can the authors calculate the enrichment efficiency from the cTnI spike-in serum samples?

7. In figure S13, the unfunctionalized NPs(i) have very poor enrichment performance. Is that because of the hydrophobicity of NP-BAPTES before peptide conjugation? The NPs conjugated with control peptide(iii) showed much weaker enrichment of cTnI but similar enrichment of nonspecific proteins compared with NPs conjugated with targeting peptide. Could the authors compare the accumulation percentage of the 10 most abundant proteins (loading mixture)? This may help readers understand how to eliminate the adsorption of nonspecific proteins.

8. It is very impressive that targeting peptide-conjugated NPs have an excellent capacity to deplete human serum albumin, but the authors did not mention whether the control peptide-conjugated NPs also have this ability. Where does the albumin-depletion property come from? The NP itself or the peptide sequence? Can the authors compare the albumin depletion capability of targeting peptide-conjugated NPs and control peptide-conjugated NPs?

9. SDS-PAGE gel shows obvious protein enrichment, including the target cTnI. Since targeting peptides can specifically and tightly bind with cTnI, could the authors perform a second elution to determine whether some cTnI and other proteins are left on the surface of NPs after the first elution?

10. Stability is critical when considering the commercialization of this nanoproteomics technology. Will this NP-targeting peptide maintain its targeting performance after 3 months? 6 months?

Reviewer #2 (Remarks to the Author):

Tiambeng et al. present an interesting antibody-free approach for characterization of low abundance cardiac troponin I proteoforms directly from human serum based on a powerful combination of top-down mass spec and nanotechnology (peptide-functionalized superparamagnetic nanoparticles). The novelty is primarily in enrichment protocol, with other methods employed, including top-down mass spectrometry, being generic. The study is well planned and executed, and technical quality is high. This is however a highly targeted approach that will be of limited interest to broad audience at least in its current version. I would strongly suggest expanding the discussion of general applicability and potential for the future.

Specific comments:

Pages 4-6, Figure 1: The idea of using allenamides for selective conjugation to Cys is interesting and novel (even though this chemistry has been around for some time). It appears that the site selectivity would be lost if the peptide to be conjugated has Cys as part of the binding sequence.

Azides and cyclooctynes would be alternatives that could easily be incorporated via SPPS and be fully biorthogonal. Authors should discuss these limitations and alternative approaches.

Pages 4-6, Figure 1: Another concern is the use of thermogravimetric analysis to determine a surface density of 1.5 peptide/nm². That is an unusual way of measuring peptide loading and it is not clear from the manuscript how the authors arrived at that value. Since the peptide has two Trp residue and one Tyr, couldn't they just measure the depletion of peptide during the NP-BAPTES reaction using 280 nm absorbance and derive a more useful value such as μmol peptide per mg nanoparticle? That way at least it would be known how much total peptide ligand is being added to the sarcomeric protein extracts.

Pages 4-6, Figure 1: Because NP-Ctrl and NP-Pep contain reactive sites that have not been quenched with an exogenous thiol (e.g. glutathione or Cysteine) after the conjugation reaction, it is possible that BAPTES could react with Cys-containing proteins or other compounds downstream. Please comment.

Figure 1: The DCM/water biphasic analysis is interesting. Did the authors perform a mock reaction first with NP-BAPTES or added it after being vacuum dried (in which case it is not a fair comparison with NP-Pep).

Pages 8-9, Figure S12: The SDS-PAGE with SYPRO showed depletion of nonbinding proteins through loading, flow through, and eluate. However, the enrichment could have been better demonstrated with ELISA instead of MS peak abundance.

Page 8: "NP-BAPTES functionalized with a negative-control peptide containing alanine substitutions to reduce cTnI-binding affinity" - however there are more amino acid substitutions to the control peptide than just alanine. It is unclear what was the rationale for the other changes.

Page 9, Figure 2, Figure S12, Table S3 (and elsewhere): It appears proteoforms were identified using only accurate intact mass measurements on a Q-TOF. While this approach allows for increased sensitivity, it obviously has limitations and would have to be supplemented by MS/MS to e.g. inform on the site of phosphorylation (other PTMs). Did authors attempt to perform MS/MS measurements? Did they detect any novel proteoforms in comparison to prior art? Some discussion around these topics would have been helpful.

Page 12, Figure 3: Comparing the nanoparticles to agarose and NP-Pep to Agarose-mAb is not a fair comparison. A better comparison would have been dynabeads linked to the mAb or peptide. That would strengthen the argument that nanoparticles are superior other particles (page 3), and that the peptide is better than the mAb.

Tables S3 and S4: ppcTnI[1-206] should be ppcTnI[1-207] based on the listed observed mass. This would be consistent with Figure S12

Page 15, line 319: should it be 0.006 ng/mL (instead of 0.06 ng/mL)?

Figure S21 shows the amount of cTnI as a function of concentration (ng/mL) except in (d) where it's portrayed as amount (ng). The legend mentions loading as a function of concentration as well, but without knowing the volume loaded it's not possible to derive the estimated total cTnI loaded onto

the LC column. A table shows the calculated and observed masses of ppcTnI as 24063.7 Da, however these masses are inconsistent with (b) of the same figure and the tables in the supplemental.

Figures S22-23: The 32+ charge state of ppcTnI portrayed in figure S22 is inconsistent with the 32+ charge state portrayed in figure S23, although it appears figure 22 has the correct m/z. The entire m/z axis of figure 23 is confusing as it skips between 0.5 m/z and 5 m/z steps.

Reviewer #3 (Remarks to the Author):

Thanks for the opportunity to review this methodological manuscript. The paper is well-written, well-organized, well-illustrated, properly referenced and novel.

In this work, a multidisciplinary team of chemists, and cell & molecular biologists, presents a nanoparticle-based preparatory method for selectively, sensitively and consistently detecting an exemplary protein of clinical interest, cardiac Troponin I (cTnI).

The premise of the work relates to the longtime-, well-known- problem of plasma or serum proteome assessment that is due to the large dynamic range of proteins in terms of concentration and the dominance of the measurable proteome by such high molecular weight entities as circulating albumin. Regardless of our awareness of this issue, approaches thus far have not solved the problem.

The improvement of pre-MS preparation, beyond the use of immuno-based techniques for selection of certain proteins has long been needed. Thus, the nano-proteomic strategy to detect and quantitate proteoforms like those of cTnI is of considerable interest and potential.

My assessment of the technology is very high level. It appears logical and valid. The experimental replicates and the reliance on three different types of human heart muscle samples allows comparison of normal with disease state tissues using this methodological workflow.

The figures illustrate the nature of each experiment well, interpretable by a non-expert.

A few questions which if answered and with answers intercalated into the paper should add a little value; they are as follows:

1. Is it assumed that the approach used for cTNI would/will work for the other troponins? Will it work for other low abundance proteins in plasma or serum like cytokines, growth factors, etc.? Please elaborate the basis of this belief? If other low abundance proteoforms could be assessed, what would be the hurdles for doing so that are not covered by the work that you present here?
2. What is the practicability of the nano-proteome strategy versus immuno-strategies in terms of time, various costs and broader applicability for the detection of other low abundance proteoforms?
3. While this nano-proteomic technique appears sensitive, reproducible, etc., will it be so when plasma or serum from patients with different levels of blood lipids, blood sugar, etc., are encountered?
4. What is the value of showing the different nano-proteoforms between the three hearts that were studied? Do you have any insight as what those apparent differences might mean? Please elaborate.
5. When you do the spike-in experiments, which cTnI do you use? Why? Would a different source impact your results? Overall, how do you assure specificity of what you are measuring (realizing the problems of specificity that exist with immuno-pre-MS techniques)?
6. Will this advanced technique ever have clinical relevance? Please explain how and likely when? What are the hurdles?

Thanks for the privilege of reviewing this paper.

Reviewer Comments

Reviewer #1 (Remarks to the Author):

Top-down mass spectrometry (MS) proteomics plays a critical role in identifying post-translational modification of proteins. However, the detection and analysis of low-abundance proteins in plasma/serum by MS has proven challenging. The authors developed a nanoproteomics platform by conjugating superparamagnetic nanoparticles with targeting peptides to enrich cardiac troponin I (cTnI), a biomarker for cardiovascular disease, in serum samples. The nanoproteomics platform showed excellent cTnI enrichment relative to other abundant proteins in plasma. The authors performed a thorough demonstration of the utility of this platform, showing diverse cTnI proteoforms and establishing the proteoform-pathophysiology relationship. This is compelling work that could be further improved by addressing the following critiques before publication in Nature Communications.

Response: We are grateful to the Reviewer for the highly positive comments.

1.1. From Fig 1c-1e, it appears that there may be Fe₃O₄-peptide aggregation. Therefore, the reviewer suggests adding more experiments on the dispersibility and size uniformity of Fe₃O₄-peptide, as this may influence surface area and cTnI enrichment.

Response: We thank the Reviewer for the constructive comments. Following the Reviewer's suggestions, we have included a new **Supplementary Figure 10** (page S18, also enclosed below) showing zeta potential analysis of NP-Pep at physiological pH (7.4). We have also added an additional sentence in the main text (page 7) describing the use of zeta potential to confirm dispersibility and colloidal stability: “To demonstrate the colloidal stability of the NP-Pep, we determined the zeta potential (ζ -potential) of the NP-Pep suspended in 0.1x PBS buffer (pH 7.4) to be ~ -38 mV, which has been previously shown to be both ideal for serum protein applications and sufficient for electrostatic repulsive forces to dominate over the van der Waals force, such that agglomeration is suppressed^{1,2}”

Supplementary Figure 10

Summary of zeta potential measurements on NP-Pep.

a, Plots of apparent zeta potentials for NP-Pep suspended in 0.1x PBS buffer (pH = 7.4) at a mass loading concentration of 0.25 mg/mL. Data are representative of $n = 10$ independent NP-Pep syntheses from the same NP-BAPTES batch. **b**, Table summary of electrophoretic mobility, conductivity, and zeta potential results as shown in (a).

1.2. While SDS-PAGE provides a gross measure of assay reproducibility, it has important limitations. It would be more compelling to see bottom-up MS data of three replicates (NP-Targeting Peptide) from the serum with data presentation in a Venn diagram to show reproducibility.

Response: We thank the Reviewer for raising these concerns. To clarify, while we show SDS-PAGE as one measure of assay reproducibility, we further demonstrate the reproducibility of the NP-Pep enrichment using top-down LC/MS analysis of the exact protein mixtures that were loaded in the each SDS gel text (featured in **Fig. 2c-e** and **Fig. 3b-d**). We thank the Reviewer for suggesting bottom-up MS analysis, but we believe top-down MS is better suited to demonstrate assay reproducibility. Shotgun/bottom-up MS proteomics is not ideally suited for validating reproducibility because it suffers from issues related to variable protein digestion and irreproducible protein identification/quantitation³⁻⁵. In contrast, top-down MS proteomics has been shown to be highly quantitative and provides a reproducible method for assaying complex biological differences, even by a label-free approach⁶⁻⁸. Following the Reviewer's suggestion to further illustrate the assay's reproducibility, we have incorporated a new **Supplementary Figure 14** (page S22) showing top-down LC/MS analysis of six NP-Pep enrichment elution mixtures arising from three different inter- and intra-batch NP-Pep enrichment elution mixtures. Moreover, we have provided additional serum NP-Pep enrichment characterization by top-down LC/MS in a new **Supplementary Figure 22** (page S30). These results unambiguously demonstrated the reproducibility of this nanoproteomics assay from batch-to-batch and from sample-to-sample. Below are the specific new Supplementary Figures provided in this revision:

Supplementary Figure 14

Top-down LC/MS cTnI enrichment reproducibility by NP-Pep.

a-b, Total ion chromatogram mass spectra (TIC-MS) of six independent NP-Pep elution mixtures (E) obtained from human heart extracts. The elution mixtures are shown for each run (**a**) and overlaid in a single plot (**b**). Equal amounts of NP-Pep (5 mg) were used for cTnI enrichment containing 0.3% cTnI obtained

from a human donor heart. **c**, Raw MS1 of cTnI obtained from the NP-Pep elution mixtures corresponding to **(a-b)**. **d**, Deconvoluted mass spectra corresponding to enriched cTnI in **(a-c)**. Roman numerals correspond to N-terminally acetylated cTnI proteoforms following Met exclusion: (i) *ppcTnI*[1-207]; (ii) cTnI; (iii) *pcTnI*; (iv) *ppcTnI*. cTnI proteoforms were identified based on accurate intact mass measurement, using the most abundant mass calculated from the amino acid sequence of entry name TNNI3_human from the UniProtKB sequence database. Data correspond to the results shown in **Fig. 2** and **Supplementary Fig 13**.

Supplementary Figure 22

Top-down LC/MS serum cTnI enrichment reproducibility and characterization by NP-Pep.

a, Total ion chromatogram mass spectra (TIC-MS) of three independent NP-Pep elution mixtures (E) obtained from a serum spike-in cTnI enrichment. Equal amounts of NP-Pep (5 mg) from separate synthesis batches were used for cTnI enrichment from human serum (10 mg) containing a minimal spike-in of cTnI (final concentration of 18.7 ng/mL), obtained from a human donor heart. **b**, Raw MS1 of cTnI obtained from the NP-Pep elution mixtures corresponding to **(a)**.

1.3. Fig. 11 demonstrates the authors' claim that the salt (NaCl) concentration of the wash buffer was a critical tunable parameter to promote effective cTnI enrichment. It would be helpful to test that hypothesis and explain this phenomenon with additional relevant experiments. In addition, the authors should also determine the zeta potential of the NP-Pep since the electric charge intensity could influence cTnI enrichment when using different NaCl concentrations as the washing buffer.

Response: We thank the Reviewer for the constructive comments. Please refer to the previous response (*response 1.1*) and the new **Supplementary Figure 10** showing the zeta potential of the NP-Pep at physiological pH. For reference, the zeta potential of the NP-Pep was determined to be ~ -38 mV at pH = 7.4. Although the exact mechanism on how the salt concentration influences NP-Pep binding with cTnI is very complicated (as previous works have demonstrated)⁹⁻¹¹ and would require extensive study beyond the scope of the current work, there are previous literature reports that suggest “charge-screening” effects¹² to be a likely mechanism to why increasing salt concentrations is beneficial to cTnI enrichment.

It has been previously reported that the isoelectric point of cTnI in serum patient samples is relatively acidic (pI = 5.2-5.4)¹³ and will be negatively charged at physiological pH (7.4). Because the NP-Pep is also negatively charged at physiological pH (-38 mV), there will be ionic repulsion between the NP-Pep and the cTnI due to like-charges. In this case, increasing salt concentrations to an optimal concentration range (~ 300 mM NaCl as determined in **Supplementary Figure 12**) can serve to minimize the ionic repulsion effects and simultaneously enhance peptide-cTnI interaction. Our data are in agreement with a previous report by Huang et. al in which they found salt concentration was a critical parameter that influenced protein binding to the aptamer-modified nanoparticles¹⁴. We would like to note that there are

additional benefits to the negative surface charge of our NP-Pep for serum enrichment, because nanoparticles with neutral and negative surface charges have been previously shown to reduce the adsorption of serum proteins, thereby improving their efficacy².

1.4. When the authors sought to evaluate the cTnI enrichment performance of NP-Pep and measure cTnI enrichment, they first used ELISA to briefly determine the cTnI ratio. Does this mean that the authors should also use ELISA to determine the ratio and then use the corresponding concentration of NP-Pep for cTnI enrichment? What determines the concentration of NP-Pep used?

Response: We thank the Reviewer for the valuable comments. To clarify, although ELISA was used to determine the initial ratio or concentration of cTnI in each biological sample, ELISA calibration is not necessary prior to NP-Pep enrichment. The concentration of NP-Pep used for serum cTnI enrichment was determined by protein assay calibrations relative to the total mass of serum protein in each sample. Experimentally, we determined that 0.5 mg NP-Pep per 1 mg of human serum was sufficient for effective cTnI enrichment (**Fig. 3**). Because the NP-Pep possess a peptide surface coverage of ~ 0.034 $\mu\text{mol/mg}$ NP-Pep (revised **Supplementary Table 3**; see *response 2.2*), the amount of NP-Pep (typically 5 mg NP-Pep per 10 mg serum) used in the serum enrichments is often in excess (~ 10,000-fold mol excess of peptide relative to cTnI) with respect to the total concentration of serum cTnI (typically ≤ 50 ng/mL). Following the Reviewer's comments, we have also added a note in the **Supplementary Methods**: “Typically, a relative loading of 0.5 mg NP-Pep per 1 mg of human serum was determined to be sufficient for effective serum cTnI enrichment”.

1.5. Although the targeting peptide can specifically adsorb cTnI proteoforms with depletion of human serum albumin, there are still other nonspecific proteins on the SDS-PAGE gel images that are normal. The authors should discuss this in more depth. In addition, could the authors analyze the most abundant proteins (e.g., the top 10) and their accumulation percentages? Does the size of nonspecific proteins and the size difference between NPs and nonspecific proteins affect enrichment performance?

Response: We thank the Reviewer for the thoughtful suggestions. Following the Reviewer's comments, we have performed a thorough investigation of the adsorbed proteins on the NP-Pep following serum enrichment. We have added the new detailed analysis summarizing the proteins reproducibly captured by the NP-Pep across multiple ($n = 5$) independent NP-Pep serum enrichments as a new **Supplementary Figure 23** (page S31). To provide more context between the size of the nonspecifically bound proteins, we note that based on the new **Supplementary Figure 23** and the SDS-PAGE results (**Fig. 3a, Supplementary Figures 16,19**) our data does not suggest an immediate relationship between the size of nonspecific serum proteins and the final enrichment performance. Hu et. al.¹⁵ have previously shown that the size of iron oxide NPs could affect the accumulation of nonspecific proteins when introduced into serum and their results suggest that smaller nanoparticles (< 200 nm) accumulate less serum proteins overall. Following the Reviewer's suggestions, we have also added additional description of the nonspecifically bound proteins following NP-Pep serum enrichment to the main text (page 15): “Although the NP-Pep demonstrates highly effective HSA depletion, there are some nonspecific proteins still retained from the serum enrichment (**Fig. 3a and Supplementary Figure 19**). To investigate these coeluted proteins, we performed a detailed analysis of all top-down LC/MS proteins identified in the NP-Pep serum elution mixtures (**Supplementary Figure 23**). cTnI was demonstrated to be confidently identified and consistently captured in all NP-Pep serum enrichment trials (**Supplementary Figure 23**)”. The new **Supplementary Figure 23** is shown below:

Top 12 Most Frequently Identified Proteins Enriched from Serum

Gene Code	Protein Name	Molecular Weight (Da)	E-Value
TNNI3	Troponin I	24063.6894	3.78E-10
TNNC1	Troponin C	18498.5755	1.19E-18
LDB3	Isoform 6 of LIM domain-binding protein 3	30890.7825	2.88E-15
CRYAB	Alpha-crystallin B chain	20188.4295	7.99E-17
RS10	40S ribosomal protein S10	18941.921	1.02E-14
APOA1	Apolipoprotein A-I	11299.7765	1.68E-07
MYL3	Myosin light chain 3	21846.1696	5.07E-10
MLRV	Myosin regulatory light chain 2	18726.18	1.04E-09
H2B1D	Histone H2B type 1-D	10725.7183	9.55E-09
H2A2C	Histone H2A type 2-C	11000.1674	3.78E-15
APOE	Apolipoprotein E	34263.6597	6.71E-14
TNNT2	Isoform 11 of Troponin T	34573.645	1.42E-03

	Identified in all $n = 5$ independent serum enrichments
	Identified in $n = 4$ independent serum enrichments
	Identified in $n = 3$ independent serum enrichments

Supplementary Figure 23

Summary and illustration of proteins identified by NP-Pep following serum enrichment.

Table summarizing the top 12 most frequently identified proteins enriched by the NP-Pep from serum cTnI spike-in enrichments obtained by combining $n = 5$ independent enrichments. cTnI (TNNI3) is confidently identified in all enrichment trials. Protein E-Value score is reported for each identification.

1.6. The targeting peptide, which has an excellent binding affinity (0.27 nM), lends the iron oxide nanoparticles outstanding capacity to enrich cTnI proteoforms. In human blood samples, the authors use the spike-in cTnI samples to test their platform. With known spike-in cTnI samples, can the authors calculate the enrichment efficiency from the cTnI spike-in serum samples?

Response: We thank the Reviewer for this suggestion. Following the Reviewer's comments, we have added a new **Supplementary Figure 27** (page S35 and also enclosed below) illustrating the cTnI enrichment efficiency of the NP-Pep from tissue and cTnI spike-in serum samples. To be more precise, there are two aspects for the "enrichment efficiency": i) enrichment factor describing the ratio of the concentrations of the cTnI after and before enrichment; ii) percent recovery describing the percentage of cTnI in the original mixture that is captured and detected after the enrichment. To incorporate this new information, we have included the following additional discussion of the serum cTnI enrichment efficiency in the main text (page 16) "Additionally, we evaluated the serum cTnI enrichment performance of the NP-Pep compared to the Agarose-mAb by ELISA detection of the cTnI amount before and after enrichment (**Supplementary Fig. 27**). The NP-Pep demonstrated a high cTnI enrichment factor (115-fold) and the serum cTnI percent recovery of the NP-Pep is ~ 51% (3-fold higher than the Agarose-mAb, ~ 17%). We believe that the cTnI percent recovery can be further improved with future optimizations in automating the NP-Pep enrichment workflow to reduce sample handling and transfer steps which may result in unnecessary protein loss. Furthermore, additional instrumentation improvements in top-down MS¹⁶ will further improve the LOD of the platform toward the diagnostic cutoff value used by contemporary cTnI ELISA (≤ 0.04 ng/mL)".

Supplementary Figure 27

ELISA-based cTnI enrichment efficiency quantification of NP-Pep and Agarose-mAb.

a, ELISA-based colorimetric quantification of cTnI standards (blue dashed box) and enrichment samples (black dashed box). All samples were dispensed in triplicate and the assay was performed according to the manufacturer's instructions. The ELISA assay uses capture antibodies targeting cTnI amino acids 18-28 and 86-90, with detection antibodies targeting cTnI amino acids 41-49. **b**, ELISA-based standard curve (0 ng, 0.4 ng, 1.25 ng, 2.5 ng, 7.5 ng, 20 ng) used for the quantification of cTnI amount before enrichment and after enrichment by NP-Pep or Agarose-mAb. **c**, Summary of enrichment performance results for NP-Pep (tissue/serum) and Agarose-mAb (serum).

$$\text{cTnI enrichment factor} = \frac{\text{concentration cTnI After Enrichment } \left(\frac{\text{ng}}{\text{mL}}\right)}{\text{concentration cTnI Before Enrichment } \left(\frac{\text{ng}}{\text{mL}}\right)}$$

$$\text{cTnI percent recovery} = \frac{\text{cTnI After Enrichment (ng)}}{\text{cTnI Before Enrichment (ng)}} \times 100\%.$$

1.7. In figure S13, the unfunctionalized NPs(i) have very poor enrichment performance. Is that because of the hydrophobicity of NP-BAPTES before peptide conjugation? The NPs conjugated with control peptide(iii) showed much weaker enrichment of cTnI but similar enrichment of nonspecific proteins compared with NPs conjugated with targeting peptide. Could the authors compare the accumulation percentage of the 10 most abundant proteins (loading mixture)? This may help readers understand how to eliminate the adsorption of nonspecific proteins.

Response: We appreciate the Reviewer's comments. The most important reason for the poor enrichment performance of unfunctionalized NPs is that they do not have any functionalized groups that can specifically recognize and bind to the cTnI. We also agree with the Reviewer that the hydrophobicity of NP-BAPTES before peptide conjugation also contribute to the poor enrichment performance of unfunctionalized NPs. As the Reviewer suggested, we have included a new **Supplementary Figure 23** (refer to the previous response 1.5) detailing the accumulation of proteins on the NP-Pep during protein enrichment. Referring to **Supplementary Figure 13 (now labeled as Supplementary Fig. 15 in the revised Supplementary Information)**. We believe that the similarity of the nonspecifically bound proteins between the NPs conjugated with control peptide (iii) and the final NP-Pep (iv) is likely a reflection of the similar general

physiochemical states of these NPs and the most abundant nonspecific proteins found from the protein loading mixture (L). On the other hand, the physiochemical states of the unfunctionalized NPs (i) (**Supplementary Figure 15**) are very different from those of the peptide functionalized NPs, and they display minimal nonspecific protein binding. As suggested by the Reviewer, we do suspect that the resistance to nonspecific proteins may likely be a combination of ionic and hydrophobic effects, because the major classes of interactions of proteins in aqueous solution involve ionic/electrostatic, hydrophobic/entropic, and H-bonding interactions¹⁷. Future works will aim to better understand these protein adsorption behaviors to better rationally design surface functionalization chemical motifs to minimize the absorption of nonspecific proteins for protein enrichment from complex biological mixtures.

1.8. It is very impressive that targeting peptide-conjugated NPs have an excellent capacity to deplete human serum albumin, but the authors did not mention whether the control peptide-conjugated NPs also have this ability. Where does the albumin-depletion property come from? The NP itself or the peptide sequence? Can the authors compare the albumin depletion capability of targeting peptide-conjugated NPs and control peptide-conjugated NPs?

Response: We thank the Reviewer for the constructive comments. From our general experience, the control peptide-conjugated NPs also showed a significant depletion of human serum albumin, although the nonspecific binding was similar to the final NP-Pep. We did not specifically investigate the human serum albumin depletion by the control peptide-conjugated NPs in detail, because they only served to mostly illustrate the critical role of the specific peptide for the final NP-Pep formulation. To comment on the origin of the albumin depletion property, we note that the agarose-beads functionalized with the same cTnI-targeting peptide (Agarose-Pep) demonstrated significant albumin retention (**Fig. 3, Supplementary Figures 19-21**). On the other hand, the unfunctionalized BAPTES-NPs (NP-Ctrl) displayed minimal nonspecific protein binding and similar depletion of HSA, compared to the final NP-Pep (**Fig. 3, Supplementary Figures 19**). These results imply that the NP itself is responsible for the albumin-depletion property, not the peptide sequence.

1.9. SDS-PAGE gel shows obvious protein enrichment, including the target cTnI. Since targeting peptides can specifically and tightly bind with cTnI, could the authors perform a second elution to determine whether some cTnI and other proteins are left on the surface of NPs after the first elution?

Response: We thank the Reviewer for this suggestion. We actually had previously attempted subsequent elutions using 200 mM glycine hydrochloride, pH 2.2, or 1% SDS and we found that the majority of proteins were released exclusively in the first elution mixture, with minimal protein amounts detected in subsequent elutions. Because of this, our routine procedure only uses one elution. We have added a note about this in the **Supplementary Methods**.

1.10. Stability is critical when considering the commercialization of this nanoproteomics technology. Will this NP-targeting peptide maintain its targeting performance after 3 months? 6 months?

Response: We agree that stability will be critical for the future commercialization of this nanoproteomics technology and we will continue to investigate shelf-life and storage conditions in future studies. Currently, we have preliminary estimates that the NP-Pep remains stable when suspended in aqueous media at 4 °C for at least 3 months, which was the longest period of time we had kept using one batch of fully functionalized NP-Pep so far. Our future efforts will include accelerated stability testing to understand short- and long-term NP-Pep stability, which would be important for efforts toward commercialization of this technology. We have already filed a provisional patent (Ge Y.; Jin, S; Tiambeng T. N., Roberts D. S. “Accurate and Comprehensive Cardiac Troponin I Assay Enabled by Nanotechnology and Proteomics” Provisional Patent 62/949,869 filed December 18, 2019).

Reviewer #2 (Remarks to the Author):

Tiambeng et al. present an interesting antibody-free approach for characterization of low abundance cardiac troponin I proteoforms directly from human serum based on a powerful combination of top-down mass spec and nanotechnology (peptide-functionalized superparamagnetic nanoparticles). The novelty is primarily in enrichment protocol, with other methods employed, including top-down mass spectrometry, being generic. The study is well planned and executed, and technical quality is high. This is however a highly targeted approach that will be of limited interest to broad audience at least in its current version. I would strongly suggest expanding the discussion of general applicability and potential for the future.

Response: We thank the Reviewer for the critical comments. To expand on the discussion of general applicability and potential for the future we have revised the main text and included new points in the conclusions section (pages 19-20): “*This antibody-free approach can be leveraged in future clinical cTnI diagnostic assays. By further applying to a large human cohort, patient blood samples can be analyzed to comprehensively detect all cTnI proteoforms and establish the relationships between cTnI proteoforms and underlying disease etiology¹⁸*” and “*Ultimately, this nanoproteomics strategy could enable next-generation precision medicine approaches for comprehensive cTnI analysis toward accurate diagnosis, better risk stratification, and improved outcome assessment of patients presenting with various cardiovascular syndromes. Beyond cTnI, we expect that this scalable and reproducible top-down nanoproteomics approach can be generally applied to other low-abundance plasma/serum proteins of interest, such as cytokines or growth factors, when provided if the NPs can be functionalized with a suitable affinity reagent^{19,20}. With the exciting recent advances in phage display libraries²¹, *in silico* techniques²², and designing aptamers/affimers have made largely owing to advancements in high-throughput methods for systemic evolution of ligands by exponential enrichment (SELEX)^{23,24}, the design and selection of effective affinity reagents for targeted protein analysis is more accessible than ever before. This nanoproteomics strategy is capable of providing previously unachievable molecular details of low-abundance serum proteins in general and can serve as an enabling technology to comprehensively map the proteoform landscape.*”

2.1. Specific comments:

Pages 4-6, Figure 1: The idea of using allenamides for selective conjugation to Cys is interesting and novel (even though this chemistry has been around for some time). It appears that the site selectivity would be lost if the peptide to be conjugated has Cys as part of the binding sequence. Azides and cyclooctynes would be alternatives that could easily be incorporated via SPPS and be fully biorthogonal. Authors should discuss these limitations and alternative approaches.

Response: We thank the Reviewer for the excellent suggestion. First, we would like to clarify that the cysteine (Cys) amino acid used here for selective peptide conjugation to the nanoparticles is not part of the original binding sequence (HWQIAYNEHQWQ) and is instead further appended to the C-terminus of the peptide (HWQIAYNEHQWQ-Cys) for conjugation. Additionally, we have previously tested a similar peptide sequence with a Gly-Gly-Gly spacing linker between the 12-mer cTnI-peptide binding sequence and the appended cysteine residue, with no noticeable difference in resultant cTnI enrichment performance. However, we agree that the current allenamide method coupling strategy requires available Cys residues that not critical to the binding sequence. Alternative coupling methods, such as the mentioned azides and cyclooctynes, can be promising alternatives in the case a nonessential Cys residue is not available. Following the Reviewer’s suggestions, we have added an additional sentence in the main text (page 5) discussing alternative strategies for biorthogonal conjugation approaches: “*It should be noted that such allene carboxamide chemistry relies on the presence of a Cys nonessential to the peptide binding sequence. In the case where such a Cys is not available, alternative bioorthogonal coupling approaches, such as azides and cyclooctynes, can be employed^{25,26}*”.

2.2. Pages 4-6, Figure 1: Another concern is the use of thermogravimetric analysis to determine a surface density of 1.5 peptide/nm². That is an unusual way of measuring peptide loading and it is not clear from the manuscript how the authors arrived at that value. Since the peptide has two Trp residue and one Tyr,

couldn't they just measure the depletion of peptide during the NP-BAPTES reaction using 280 nm absorbance and derive a more useful value such as μmol peptide per mg nanoparticle? That way at least it would be known how much total peptide ligand is being added to the sarcomeric protein extracts. The calculation of surface density is derived from normalized weight loss Measurement at 280 nm could potentially be another way of verifying the concentration.

Response: We thank the Reviewer for the constructive comments. While UV-Vis measurement at 280 nm is typical and would be convenient for measuring peptide concentration, the iron oxide nanoparticles show a strong sloping absorption feature throughout the visible range (200 - 600 nm) that interferes with potential UV-Vis measurements, especially at $\sim 280\text{ nm}$ ²⁷. In this case, UV-Vis is not well-suited for quantifying surface peptide coverage. Actually, the use of thermogravimetric analysis (TGA) to quantify nanoparticle surface organic coating is a common and well-established analytical technique and has been previously used to quantify surface biomolecules, such as peptides, on nanomaterials²⁸⁻³¹.

But we agree with the Reviewer that the TGA determination of peptide surface density should be described in more detail and we thank the Reviewer for the suggestion to report the surface density as μmol peptide per mg nanoparticle as a more informative metric. Moreover, we would like to thank the Reviewer for raising the concern, as we have found an arithmetic error during the calculation of the peptide surface density (previously reported as 1.5 peptide/nm²). The corrected peptide surface density (0.2 peptide/nm² or 0.034 μmol peptide per mg nanoparticle) has been revised in the main text (page 7) with the following sentence: “From the difference in weight loss ($\sim 6\%$) between the NP-BAPTES and the final NP-Pep, a surface density of $\sim 0.034\ \mu\text{mol}$ peptide/mg NP was inferred (Supplementary Table 1-3)”. Following the Reviewer’s suggestion, we have incorporated the mentioned changes in a revised Supplementary Table 1 (page S39) and two new Supplementary Tables 2-3 (pages S40-41 and also shown below) that detail the surface coverage analysis by TGA.

Supplementary Table 2. Summary showing the details of using TGA analysis to estimate the surface density of BAPTES ligands on the surface of NPs.

Nanoparticle	NP-BAPTES
Average radius (obtained from TEM)	4.0 nm
The mass of BAPTES ligand molecules (quantified by TGA % mass loss), m	$26\% \times 4.313\text{ mg}$ (Sample size used for TGA) = 1.121 mg BAPTES
Total number of BAPTES ligand molecules = $\frac{m}{MW} \times N_A$ where MW , ligand molecular weight; N_A , Avogadro’s number ($6.022 \times 10^{20} \frac{\text{molecules}}{\text{mmol}}$)	$\frac{1.121\text{ mg BAPTES}}{287.12 \frac{\text{mg}}{\text{mmol}}} \times 6.022 \times 10^{20} \frac{\text{molecules}}{\text{mmol}}$ = $\sim 2.35 \times 10^{18}$ BAPTES ligand molecules
The total mass (M) of NPs in TGA sample	$4.313\text{ mg sample} - 1.207\text{ mg BAPTES}$ = 3.192 mg NPs
The total volume (V) of NPs from TGA sample ($\rho = \frac{5.18\text{ g}}{\text{cm}^3}$)	$V = M / \rho$; $6.162 \times 10^{-4}\text{ cm}^3$
The volume (v) of a single NP	$v = \frac{4}{3} \pi (4 \times 10^{-7}\text{ cm})^3 = 2.68 \times 10^{-19}\text{ cm}^3$
Total # of NPs from TGA sample:	$\frac{V}{v} = \frac{6.95 \times 10^{-4}\text{ cm}^3}{2.68 \times 10^{-19}\text{ cm}^3} = 2.298 \times 10^{15}$ NPs
# of BAPTES ligands per NP	~ 1023
SA (total surface area) $N \times 4\pi r^2$	$4.622 \times 10^{17}\text{ nm}^2$
# of BAPTES ligands per nm ²	~ 5
μmol BAPTES ligands per mg of nanoparticle	$\sim 0.90\ \mu\text{mol}$ BAPTES/mg of NP

Supplementary Table 3. Summary showing the details of using TGA analysis to estimate the surface density of cTnI peptide ligands on the surface of NPs.

Nanoparticle	NP-Pep
Average radius (obtained from TEM)	4.0 nm
The mass of BAPTES and peptide ligand molecules (quantified by total TGA % mass loss)	$32\% \times 2.434 \text{ mg}$ (Sample size used for TGA) = 0.779 mg BAPTES + Peptide ligand
The mass of peptide ligand molecules (quantified by TGA % mass loss (i.e. 26%) compared to NP-BAPTES), m	$6\% \times 2.434 \text{ mg}$ (Sample size used for TGA) = 0.146 mg peptide
Total number of peptide ligand molecules = $\frac{m}{MW} \times N_A$ where MW , ligand molecular weight; N_A , Avogadro's number ($6.022 \times 10^{20} \frac{\text{molecules}}{\text{mmol}}$)	$\frac{0.146 \text{ mg peptide}}{1742.89 \frac{\text{mg}}{\text{mmol}}} \times 6.022 \times 10^{20} \frac{\text{molecules}}{\text{mmol}}$ = $\sim 5.046 \times 10^{16}$ peptide molecules
The total mass (M) of NPs in TGA sample	$2.434 \text{ mg sample} - 0.779 \text{ mg BAPTES} + \text{peptide}$ = 1.655 mg NPs
The total volume (V) of NPs from TGA sample ($\rho = \frac{5.18 \text{ g}}{\text{cm}^3}$)	$V = M/\rho$; $3.195 \times 10^{-4} \text{ cm}^3$
The volume (v) of a single NP	$v = \frac{4}{3} \pi (4 \times 10^{-7} \text{ cm})^3 = 2.68 \times 10^{-19} \text{ cm}^3$
Total # of particles from TGA sample:	$\frac{V}{v} = \frac{3.197 \times 10^{-4} \text{ cm}^3}{2.68 \times 10^{-19} \text{ cm}^3} = 1.191 \times 10^{15}$ NPs
# of peptide ligands per NP	~ 42
SA (total surface area) $N \times 4\pi r^2$	$2.396 \times 10^{17} \text{ nm}^2$
# of peptide ligands per nm^2	~ 0.2
μmol peptide ligands per mg of nanoparticle	$\sim 0.034 \mu\text{mol}$ peptide/mg of NP

2.3. Pages 4-6, Figure 1: Because NP-Ctrl and NP-Pep contain reactive sites that have not been quenched with an exogenous thiol (e.g. glutathione or Cysteine) after the conjugation reaction, it is possible that BAPTES could react with Cys-containing proteins or other compounds downstream. Please comment.

Response: We thank the Reviewer for raising this concern. We would like to clarify that the peptide coupling reaction with BAPTES via the allene carboxamide chemistry requires elevated pH (8.0-9.0) for effective coupling. In the enrichment experiments performed at buffered physiological pH, the efficiency for potentially coupling to Cys-containing proteins or other Cys-containing compounds should be reduced, so this should not be a significant concern. Furthermore, our data did not suggest that nonspecific protein binding or reduced cTnI enrichment performance were a result of unquenched reactive sites, as the NP-Ctrl bound minimal proteins in all enrichment trials. As shown in **Figs. 2-3** and **Supplementary Figures 15-16** and **18-21**, the NP-Ctrl and NP-Pep demonstrate good resistance to nonspecific protein binding in both tissue and serum protein mixtures, without requiring additional quenching steps. However, we do thank the Reviewer for the excellent suggestion, and we believe that future studies can further explore additional capping or quenching of unbound reactive sites to reveal their effects on downstream enrichment performance or nonspecific protein accumulation.

2.4. *Figure 1: The DCM/water biphasic analysis is interesting. Did the authors perform a mock reaction first with NP-BAPTES or added it after being vacuum dried (in which case it is not a fair comparison with NP-Pep).*

Response: To clarify the dichloromethane/water biphasic photos, the coupling reaction with peptide was not carried out *in situ* in such biphasic mixture. The intent is to contrast the dispersibility of the two type of NPs (NP-BAPTES and NP-Pep) side-by-side. Using the same fresh NP-BAPTES batch, one set of the as-synthesized NP-BAPTES was placed in the DCM/water biphasic mixture (the vial on the left) and the other set with equal amount of NPs was separately peptide-coupled then added into another vial containing the DCM/water biphasic mixture (shown for the vial on the right). Now we realize that there could be misunderstanding and confusion about these comparison photos, therefore, we have revised **Fig. 1h** to remove the “peptide coupling/1h” arrow in the middle, and revised the caption of **Fig. 1h** in the main text (page 6): “Photographs of functionalized NPs in a biphasic mixture of dichloromethane (CH₂Cl₂) and water (H₂O), comparing the solvent compatibility of the NP-BAPTES and the NP-Pep. The NP-BAPTES are dispersible in dichloromethane but the NP-Pep are stable and dispersible in water. The displayed NP-Pep and NP-BAPTES originated from the same synthetic batch.”

2.5. *Pages 8-9, Figure S12: The SDS-PAGE with SYPRO showed depletion of nonbinding proteins through loading, flow through, and eluate. However, the enrichment could have been better demonstrated with ELISA instead of MS peak abundance.*

Response: We thank the Reviewer for the constructive comments. We have additionally demonstrated the enrichment using ELISA (new **Supplementary Figure 27**, page S35) and provided new information on the enrichment (*response 1.6*). Importantly, we would like to clarify that quantitative enrichment performance analysis can indeed be performed by top-down LC/MS. Lin et. al.⁶ have previously demonstrated that top-down LC/MS can be used to reliably quantify protein expression and relative protein concentrations across different samples by extracted ion chromatogram (EIC) analysis when normalizing LC/MS sample loading to the total protein amount present in a mixture (*e.g.*, 500 ng in the current example, **Supplementary Methods**). We would like to also note that by top-down LC/MS analysis, we show simultaneous quantification of cTnI relative abundance with additional molecular insights to changes in relative proteoform levels as well as any protein post-translational modifications (PTMs), which is not possible with ELISA. Furthermore, we would like to emphasize that while ELISA is useful at quantifying the amount of cTnI before and after enrichment, the SDS-PAGE and top-down LC/MS analysis reveal additional important information related to global protein abundance changes and highlight the impressive resistance to nonspecific serum proteins (such as human serum albumin; HSA) that these NP-Pep feature (**Fig. 3** and **Supplementary Figures 18-21**).

2.6. *Page 8: “NP-BAPTES functionalized with a negative-control peptide containing alanine substitutions to reduce cTnI-binding affinity” - however there are more amino acid substitutions to the control peptide than just alanine. It is unclear what was the rationale for the other changes.*

Response: We thank the Reviewer for raising this concern. Our selection of this ‘negative-control’ peptide sequence is taken from the work by Xiao et. al., in which the authors demonstrate that this particular ‘negative-control’ sequence yielded diminished affinity towards cTnI²². We agree with the Reviewer’s point that this negative-control peptide sequence contains residue changes beyond just alanine substitutions. Although not explicitly detailed in the previous study, the additional residue changes may have been made to alter the secondary structure of the negative-control sequence to have less favorable interactions with cTnI. We have clarified the description of the negative-control peptide in the main text (page 8) and in **Supplementary Figure 13** by removing the mention of just “alanine substitution” and instead referencing the previous study, as follows: “NP-BAPTES functionalized with a negative-control peptide²² to reduce cTnI-binding affinity”.

2.7. Page 9, Figure 2, Figure S12, Table S3 (and elsewhere): It appears proteoforms were identified using only accurate intact mass measurements on a Q-TOF. While this approach allows for increased sensitivity, it obviously has limitations and would have to be supplemented by MS/MS to e.g. inform on the site of phosphorylation (other PTMs). Did authors attempt to perform MS/MS measurements? Did they detect any novel proteoforms in comparison to prior art? Some discussion around these topics would have been helpful.

Response: We thank the Reviewer for raising this concern. Following the Reviewer's suggestions, we have performed MS/MS analysis on the detected cTnI proteoforms and we have included a summary of the analysis as a new **Supplementary Figure 29** (page S37, also reproduced below). We have also added additional details of the MS/MS analysis in the main text (page 17): "Tandem MS/MS analysis of the detected serum cTnI proteoforms were used to validate proteoform assignments across the various heart pathologies (Supplementary Figure 29)". To clarify, we did not detect any novel cTnI proteoforms in comparison to prior art. The current work did not seek to reveal new cTnI PTMs, rather we aimed to develop a new technology capable of enriching cTnI from serum while globally preserving all endogenous cTnI proteoforms and their relative abundances with no artifactual modifications. This proteoform-resolved technology holds promise for potentially revealing new cTnI PTMs or establishing previously unknown proteoform-pathophysiology relationships cTnI proteoforms in future studies. The new **Supplementary Figure 29** is shown below:

Supplementary Figure 29

Top-down LC-MS/MS characterization of cTnI arising from the various biological samples after enrichment.

a-c, Representative CID fragment ions obtained from cTnI arising from the donor heart (y_{76}^{13+} , y_{209}^{30+} , b_{86}^{12+} , and b_{155}^{21+}), the diseased heart (y_{76}^{12+} , y_{42}^{7+} , b_{31}^{5+} , and b_{86}^{12+}), and the post-mortem heart (y_{76}^{8+} , y_{51}^{9+} , b_{86}^{14+} , and b_{31}^{5+}) sources. cTnI was found to be primarily in its bis-phosphorylated state in the donor heart (ppcTnI; Ser22, and Ser23), unphosphorylated state in the diseased heart (cTnI), and in a proteolytically degraded form in the post-mortem heart (cTnI[1-206]). Theoretical ion distributions are indicated by the red dots and mass accuracy errors are listed for each fragment ion. **d-f**, Protein sequence fragmentation mapping of the specific proteoform of cTnI corresponding to the fragment ion data obtained from each cTnI source (**a-c**). CID fragmentations are shown as red cleavages. All matched sequences contained N-terminally acetylated cTnI proteoforms following Met exclusion. Amino acid sequence was based on the entry name TNNI3_human obtained from the UniProtKB sequence database.

2.8. Page 12, Figure 3: Comparing the nanoparticles to agarose and NP-Pep to Agarose-mAb is not a fair comparison. A better comparison would have been dynabeads linked to the mAb or peptide. That would strengthen the argument that nanoparticles are superior other particles (page 3), and that the peptide is better than the mAb.

Response: We thank the Reviewer for the critical comments. While we did not test the difference between our nanoparticles *versus* Dynabeads in the current work, our choice to compare the NP-Pep to the conventional agarose-based platform was due to agarose's wide commercial availability and its well-established history for general affinity purification for biological systems³². For the purposes of the current study, we believe that the comparison to the agarose platform is useful as a baseline metric of enrichment performance and is informative to demonstrating the general utility of the NP-Pep platform. Moreover, by coupling the same cTnI-binding peptide as well as additional cTnI-binding mAb to the agarose platform (Agarose-Pep and Agarose-mAb, respectively), we were able to examine the effects of affinity ligand choice and material choice on enrichment performance when compared to our surface-functionalized nanoparticles.

Our tissue and serum enrichment data demonstrated that the Agarose-Pep and Agarose-mAb yielded similar cTnI enrichment performance (**Fig. 3** and **Supplementary Figures 17-19**). Our major finding was that only the nanoparticles surface-functionalized with the newly synthesized BAPTES molecule demonstrated impressive resistance to nonspecific binding of highly abundant serum proteins, which was not found to be due to the choice of affinity reagent. In this case, we feel that the aforementioned comparison of the nanoparticles to the agarose platform is fair. For reference, Thermo's Dynabeads also compare to agarose platforms as a benchmark for their immunoprecipitation products and they recommend pre-blocking with bovine serum albumin (BSA) or addition of non-ionic surfactants (such as Tween 20 or Triton X-100) to reduce nonspecific protein binding (<https://www.thermofisher.com/us/en/home/life-science/protein-biology/protein-assays-analysis/immunoprecipitation/immunoprecipitation-faqs-dynabeads-magnetic-beads.html#6>). Our NP-Pep neither requires neither the use of surfactants, which can significantly suppress protein MS signal³³⁻³⁵, nor any pretreatment with other additives, such as BSA blocking, to enable its impressive resistance to nonspecific binding. Although enrichment performance comparisons to other particle systems may also be informative and can potentially further reveal the performance of the NP-Pep system across multiple platforms, we feel that this is not essential for publication especially during the current COVID-19 pandemic.

To further clarify, we did not argue that the "peptide is better than the mAb" for cTnI enrichment performance. However, there are distinct advantages to the peptide-based approach that our data highlight: (1) similar enrichment performance was seen between the Agarose-mAb and Agarose-Pep platforms in both tissue and human serum enrichment experiments; (2) the peptide is more reproducible because it is a short-chained molecule as opposed to an intact protein (mAb); (3) short peptides are more economical than mAbs. This current work exploits such beneficial features of this peptide to introduce a new technology for dramatically improving the capture and detection of proteins directly from serum, beyond the use of conventional immuno-based techniques.

Additionally, we have toned down the statements on the utility of nanoparticles on page 3 and have replaced "ideal" with "highly effective", as follows: "Nanoparticles (NPs) are highly effective for such sensitive and specific proteoform enrichment because: ...". Furthermore, we have changed the sentence of "Thus, these NPs can serve as antibody replacements ... in general" in the conclusions section of the main text (page 19) to the following: "Thus, these NPs can serve as replacements to conventional immuno-based techniques ... in general". We believe that these changes should justify our comparisons of the NP-Pep to the agarose-platform and improve the clarity of the manuscript overall.

2.9. Tables S3 and S4: ppcTnI[1-206] should be ppcTnI[1-207] based on the listed observed mass. This would be consistent with Figure S12.

Response: We thank the reviewer for the careful reading. We have revised the caption on **Supplementary Tables 3,4** (now **Supplementary Tables 5** and **6**, respectively) to be consistent with **Supplementary Figure 12** (now **Supplementary Figure 13**).

2.10. Page15, line 319: should it be 0.006 ng/mL (instead of 0.06 ng/mL)?

Response: We thank the Reviewer for raising the concern. Although our data shows that the mass spectrometer is capable of reliable detection of cTnI as low as 0.006 ng/mL, the number 0.06 ng/mL was

obtained by a LOD analysis using a $3.3 \sigma/s$ cutoff. To further clarify, we have revised the sentence in the main text (page 16), as follows: “Top-down RPLC/MS with a CaptiveSpray (CS) ionization source fitted to a maXis II ETD mass spectrometer was sufficiently sensitive to detect cTnI with a LOD ($3.3 \sigma/s$) as low as 0.06 ng/mL (Fig. 3e, Supplementary Fig. 25)”.

2.11. Figure S21 shows the amount of cTnI as a function of concentration (ng/mL) except in (d) where it's portrayed as amount (ng). The legend mentions loading as a function of concentration as well, but without knowing the volume loaded it's not possible to derive the estimated total cTnI loaded onto the LC column. A table shows the calculated and observed masses of ppcTnI as 24063.7 Da, however these masses are inconsistent with (b) of the same figure and the tables in the supplemental.

Response: We thank the reviewer for this comment. We have corrected the labeling in panel (c) and table (d) of **Supplementary Figure 21** (now **Supplementary Figure 25**) to be “cTnI Amount (ng/mL)”. To clarify, **Supplementary Figure 25** reports masses as the monoisotopic masses of ppcTnI. In this case, the table (d) the panel (b) are consistent. In the supplemental tables, we reported ppcTnI using most abundant masses and this report is consistent with the specific data presented in **Fig. 2e**.

2.12. Figures S22-23: The 32+ charge state of ppcTnI portrayed in figure S22 is inconsistent with the 32+ charge state portrayed in figure S23, although it appears figure 22 has the correct m/z. The entire m/z axis of figure 23 is confusing as it skips between 0.5 m/z and 5 m/z steps.

Response: We thank the reviewer for the careful reading. Moreover, we thank the Reviewer for the positive and careful evaluation, and critical but insightful comments, which helped to significantly improve this manuscript. The 32+ charge state of ppcTnI portrayed in **Supplementary Figure 22** (now **Supplementary Figure 26**) is indeed correct. The m/z axis was mislabeled on **Supplementary Figure 23** (now **Supplementary Figure 28**) and has been corrected in the newly revised **Supplementary Figure 28**.

Reviewer #3 (Remarks to the Author):

Thanks for the opportunity to review this methodological manuscript. The paper is well-written, well-organized, well-illustrated, properly referenced and novel.

In this work, a multidisciplinary team of chemists, and cell & molecular biologists, presents a nanoparticle-based preparatory method for selectively, sensitively and consistently detecting an exemplary protein of clinical interest, cardiac Troponin I (cTnI).

The premise of the work relates to the longtime-, well-known- problem of plasma or serum proteome assessment that is due to the large dynamic range of proteins in terms of concentration and the dominance of the measurable proteome by such high molecular weight entities as circulating albumin. Regardless of our awareness of this issue, approaches thus far have not solved the problem.

The improvement of pre-MS preparation, beyond the use of immuno-based techniques for selection of certain proteins has long been needed. Thus, the nano-proteomic strategy to detect and quantitate proteoforms like those of cTnI is of considerable interest and potential.

My assessment of the technology is very high level. It appears logical and valid. The experimental replicates and the reliance on three different types of human heart muscle samples allows comparison of normal with disease state tissues using this methodological workflow.

The figures illustrate the nature of each experiment well, interpretable by a non-expert.

Response: We thank the Reviewer for the extremely positive and gracious remarks.

A few questions which if answered and with answers intercalated into the paper should add a little value; they are as follows:

3.1. Is it assumed that the approach used for cTnI would/will work for the other troponins? Will it work for other low abundance proteins in plasma or serum like cytokines, growth factors, etc.? Please elaborate the basis of this belief? If other low abundance proteoforms could be assessed, what would be the hurdles for doing so that are not covered by the work that you present here?

Response: We thank the Reviewer for these comments. We expect that the described approach can be generally applied for enriching the other troponins or even other low-abundance plasma/serum proteins of interest, such as cytokines or growth factors, provided the NPs are functionalized with a suitable affinity reagent. As mentioned in our initial response to Reviewer 2 (*vide supra*), with the exciting recent advances in phase display libraries²¹, *in silico* techniques²², and the progress that aptamers/affimers have made largely owing to advancements in high-throughput methods for systemic evolution of ligands by exponential enrichment (SELEX)^{23,24}, the design and selection of high quality affinity reagents for targeted protein analysis is more accessible than ever before. An advantage of our nanoproteomics strategy is the modular nature of the nanoparticle surface-functionalization chemistry, which can allow alternative affinity reagent coupling, with slight modifications. However, there are greater challenges that involve the capture of low-abundance *proteoforms* in general. Designing effective affinity reagents that can globally capture protein-specific PTMs or isoforms has historically been challenging for the immuno-based approach³⁶. This nanoproteomics strategy represents the first platform capable of comprehensive capture and analysis of cTnI *proteoforms* with complete molecular specificity. Nevertheless, the high sensitivity of mass spectrometers still needs to be improved to achieve LODs comparable to current ELISAs. We envision the recent advances in mass spectrometry instrumentation, such as Bruker's new TIMs-TOF³⁷ and Thermo's Orbitrap Tribrid Eclipse³⁸, will help improve sensitivity. As the top-down proteomics field continues to experience its rapid growth, we anticipate a rise in new and enabling instrumentation and robust affinity reagents to address the mentioned challenges^{16,39}.

3.2. *What is the practicability of the nano-proteome strategy versus immuno-strategies in terms of time, various costs, and broader applicability for the detection of other low abundance proteoforms?*

Response: We thank the Reviewer for raising these questions. While ongoing nanoproteomics efforts are focused on enabling higher sample throughput by automation and analyzing other classes of low-abundance proteins, we believe that our nanoproteomics strategy already holds significant advantages over the traditional immuno-strategies. With regards to cost, because peptides can be synthesized at a large scale using solid phase peptide synthesis, the cost per μmol of cTnI peptide ($\sim\$4/\mu\text{mol}$ peptide; using GenScript as a specific commercial example used in this study) is significantly less than the cost per μmol of monoclonal cTnI antibody ($\sim\$200,000/\mu\text{mol}$ mAb; Santa Cruz Biotechnology chosen as specific commercial example used in this study). Additionally, we have previously demonstrated our ability to surface-functionalized iron oxide nanoparticles reproducibly and in large scales, which further aids in platform development⁴⁰. In terms of the broader applicability of the nanoproteomics strategy for the detection of other low-abundance proteoforms, please refer to our response to the previous question (3.1). We anticipate this nanoproteomics strategy will be generally applicable to the proteoform-resolved analysis of low-abundance proteins directly from serum and we will expand this nanoproteomics strategies for other low-abundance proteoforms of significant biological interest.

3.3. *While this nano-proteomic technique appears sensitive, reproducible, etc., will it be so when plasma or serum from patients with different levels of blood lipids, blood sugar, etc., are encountered?*

Response: We appreciate this important comment. As a follow-up study, we are currently developing a clinical pilot study where we will use this nanoproteomics strategy to analyze cTnI proteoforms found in clinical blood samples from patients with acute myocardial infarction compared to an apparently healthy control group. In this future work, our focus is to appreciably understand the variables of sample quality (hemolysis, lipemia) or common clinical interferents in the detection of cTnI from human blood samples using this nanoproteomics strategy.

3.4. *What is the value of showing the different nano-proteoforms between the three hearts that were studied? Do you have any insight as what those apparent differences might mean? Please elaborate.*

Response: We thank the Reviewer for this comment. The selection of the six different heart samples (which comprise a group of three specific cardiac pathologies including apparently healthy, dilated cardiomyopathic, and post-mortem) are meant to simulate the broad differences in the relative abundance of endogenous cTnI proteoforms that are likely to arise in clinical patient samples^{41,42}. Circulating cTnI has been demonstrated to exist in myriad (*e.g.*, phosphorylated, acetylated, oxidized, truncated, *etc.*) which have been shown to reflect cardiac disease status^{18,43,44}. However, immuno-based detection approaches such as ELISA are unable to distinguish these circulating proteoforms, leaving researchers devoid of reliable technologies for probing endogenous cTnI at the proteoform-resolved level.

This nanoproteomics strategy is capable of solving these challenges by sensitively enriching low-abundance cTnI proteoforms directly from human serum, while also preserving endogenous cTnI proteoform relative abundances and cTnI PTM profiles without artifactual modifications (**Fig. 2-3** and **Supplementary Figure 24**). Following the Reviewer's comments, we have added a new sentence in the main text (page 17) that details the broader significance of detecting cTnI proteoforms with respect to different and specific cardiac disease states: "**Altered PTM profiles of cTnI are associated with dysregulated cellular signaling during the onset and progression of diseases, thus disease-induced cTnI proteoforms are believed to have the potential to serve as the next generation cardiac biomarkers for diagnosis of specific cardiovascular syndromes**^{18,45,46}".

3.5. *When you do the spike-in experiments, which cTnI do you use? Why? Would a different source impact your results? Overall, how do you assure specificity of what you are measuring (realizing the problems of specificity that exist with immuno-pre-MS techniques)?*

Response: We thank the Reviewer for these important comments. To clarify, in our spike-in experiments, we used endogenous cTnI obtained from clinical human cardiac tissue samples to simulate cTnI proteoforms that may exist in human serum samples. Specifically, as mentioned in the previous response 3.4 (*vide supra*) the selection of the six different heart samples, comprising specific cardiac pathophysiology including apparently healthy (non-failing donor heart without known cardiac disease), diseased hearts (dilated cardiomyopathy), and post-mortem hearts (which provides a large number of cTnI proteoforms including phosphorylation, degradation and oxidation)⁴⁶, were chosen to simulate the rich diversity of endogenous cTnI proteoforms that are likely to arise from clinical patient samples^{41,42}. Since circulating cTnI released at the onset or during the progression of cardiac injury originates from cardiomyocytes, spike-in endogenous cTnI obtained from cardiac tissues better simulates the enrichment of endogenous cTnI proteoforms that may be found in clinical patient plasma/serum samples, as opposed to spike-in recombinant cTnI.

With regards to assuring specificity of our nanoproteomics strategy, we take advantage of top-down proteomics analysis enabled by high-resolution MS systems to provide unambiguous and highly accurate measurements of cTnI proteoforms. Such an integrated top-down MS approach holds significant advantages over existing immuno-based techniques: (1) top-down MS is capable of revealing cTnI proteoforms with total molecular specificity, providing a “bird’s eye” view of all detected proteoforms; (2) this nanoproteomics strategy, unlike existing immuno-based platforms, is highly specific due to the integration of a high specificity cTnI-binding peptide with top-down MS for highly accurate measurement, and yields a faithful and global view of diverse cTnI proteoform fingerprints arising from various PTMs of serum-enriched cTnI; (3) this nanoproteomics strategy is highly reproducible owing to its small peptide-based biorecognition element, the reproducible serum cTnI enrichment performance, and the scalable and reproducible surface-functionalized nanoparticles synthesis.

3.6. Will this advanced technique ever have clinical relevance? Please explain how and likely when? What are the hurdles?

Response: We thank the Reviewer for raising this important question. In our efforts to further the application of this technology for accurate diagnosis of cardiovascular syndromes and eventually translate this technology into the clinic, we are discussing potential collaborations with instrumentation companies such as Thermo Fisher Scientific, to develop high-throughput mass spectrometers designed specifically for sensitive cTnI detection. We are currently establishing a new clinical pilot study (n = 50 patient samples, which unfortunately was delayed due to the COVID-19 pandemic) to analyze cTnI proteoforms found in clinical patient samples with AMI compared against apparently healthy patients, with the ultimate goal of identifying cTnI proteoform biomarkers that can be further validated in a larger human cohort. While we believe that the discovery of a putative set of proteoforms can be enabled by this nanoproteomics strategy, we expect that validation of these proteoforms across a large human cohort is necessary to understand the influence of common co-morbidities and other known convoluting variables such as age and gender^{47,48}. This study is mainly focused on technology development and future efforts will be dedicated on improving sample-throughput and sensitivity to ensure that this technology becomes clinically relevant, and that mass spectrometers will become commonplace in hospitals for clinical diagnosis of cardiovascular syndromes.

Thanks for the privilege of reviewing this paper.

Response: We again thank the Reviewer for the positive and insightful comments.

References

- 1 Jiang, J., Oberdörster, G. & Biswas, P. Characterization of size, surface charge, and agglomeration state of nanoparticle dispersions for toxicological studies. *Journal of Nanoparticle Research* **11**, 77-89, doi:10.1007/s11051-008-9446-4 (2009).
- 2 Blanco, E., Shen, H. & Ferrari, M. Principles of nanoparticle design for overcoming biological barriers to drug delivery. *Nature Biotechnology* **33**, 941-951, doi:10.1038/nbt.3330 (2015).
- 3 Bell, A. W. *et al.* A HUPO test sample study reveals common problems in mass spectrometry-based proteomics. *Nature Methods* **6**, 423-430, doi:10.1038/nmeth.1333 (2009).
- 4 Tabb, D. L. *et al.* Repeatability and Reproducibility in Proteomic Identifications by Liquid Chromatography-Tandem Mass Spectrometry. *Journal of Proteome Research* **9**, 761-776, doi:10.1021/pr9006365 (2010).
- 5 Chamrad, D. & Meyer, H. E. Valid data from large-scale proteomics studies. **2**, 647-648 (2005).
- 6 Lin, Z. *et al.* Simultaneous Quantification of Protein Expression and Modifications by Top-down Targeted Proteomics: A Case of Sarcomeric Subproteome. *Molecular & Cellular Proteomics*, mcp.TIR118.001086, doi:10.1074/mcp.TIR118.001086 (2018).
- 7 Kelleher, N. L., Thomas, P. M., Ntai, I., Compton, P. D. & LeDuc, R. D. Deep and quantitative top-down proteomics in clinical and translational research. *Expert Review of Proteomics* **11**, 649-651, doi:10.1586/14789450.2014.976559 (2014).
- 8 Ntai, I. *et al.* Applying Label-Free Quantitation to Top Down Proteomics. *Analytical Chemistry* **86**, 4961-4968, doi:10.1021/ac500395k (2014).
- 9 Thanh, N. T. K. & Green, L. A. W. Functionalisation of nanoparticles for biomedical applications. *Nano Today* **5**, 213-230, doi:<https://doi.org/10.1016/j.nantod.2010.05.003> (2010).
- 10 Tsumoto, K., Ejima, D., Senczuk, A. M., Kita, Y. & Arakawa, T. Effects of salts on protein-surface interactions: applications for column chromatography. *Journal of Pharmaceutical Sciences* **96**, 1677-1690, doi:10.1002/jps.20821 (2007).
- 11 Bharti, B., Meissner, J., Klapp, S. H. L. & Findenegg, G. H. Bridging interactions of proteins with silica nanoparticles: The influence of pH, ionic strength and protein concentration. *Soft Matter* **10**, 718-728, doi:10.1039/C3SM52401A (2014).
- 12 Welsch, N., Lu, Y., Dzubiella, J. & Ballauff, M. Adsorption of proteins to functional polymeric nanoparticles. *Polymer* **54**, 2835-2849, doi:<https://doi.org/10.1016/j.polymer.2013.03.027> (2013).
- 13 Peronnet, E., Becquart, L., Martinez, J., Charrier, J.-P. & Jolivet-Reynaud, C. Isoelectric point determination of cardiac troponin I forms present in plasma from patients with myocardial infarction. *Clinica Chimica Acta* **377**, 243-247, doi:<https://doi.org/10.1016/j.cca.2006.10.006> (2007).
- 14 Huang, C.-C., Huang, Y.-F., Cao, Z., Tan, W. & Chang, H.-T. Aptamer-Modified Gold Nanoparticles for Colorimetric Determination of Platelet-Derived Growth Factors and Their Receptors. *Analytical Chemistry* **77**, 5735-5741, doi:10.1021/ac050957q (2005).
- 15 Hu, Z., Zhang, H., Zhang, Y., Wu, R. & Zou, H. Nanoparticle size matters in the formation of plasma protein coronas on Fe₃O₄ nanoparticles. *Colloids Surf B Biointerfaces* **121**, 354-361, doi:10.1016/j.colsurfb.2014.06.016 (2014).
- 16 Chen, B., Brown, K. A., Lin, Z. & Ge, Y. Top-Down Proteomics: Ready for Prime Time? *Analytical Chemistry* **90**, 110-127, doi:10.1021/acs.analchem.7b04747 (2018).
- 17 Andrade, J. D. & Hlady, V. Protein adsorption and materials biocompatibility: A tutorial review and suggested hypotheses. *Biopolymers/Non-Exclusion HPLC* **79**, 1-63 (1986).
- 18 Soetkamp, D. *et al.* The continuing evolution of cardiac troponin I biomarker analysis: from protein to proteoform. *Expert Review of Proteomics* **14**, 973-986, doi:10.1080/14789450.2017.1387054 (2017).
- 19 Dunn, M. R., Jimenez, R. M. & Chaput, J. C. Analysis of aptamer discovery and technology. *Nature Reviews Chemistry* **1**, 0076, doi:10.1038/s41570-017-0076 (2017).
- 20 Gupta, S. *et al.* Chemically Modified DNA Aptamers Bind Interleukin-6 with High Affinity and Inhibit Signaling by Blocking Its Interaction with Interleukin-6 Receptor. *Journal of Biological Chemistry* **289**, 8706-8719 (2014).

- 21 Wu, C.-H., Liu, I. J., Lu, R.-M. & Wu, H.-C. Advancement and applications of peptide phage display technology in biomedical science. *Journal of Biomedical Science* **23**, 8, doi:10.1186/s12929-016-0223-x (2016).
- 22 Xiao, X. *et al.* Advancing Peptide-Based Biorecognition Elements for Biosensors Using in-Silico Evolution. *ACS Sensors* **3**, 1024-1031, doi:10.1021/acssensors.8b00159 (2018).
- 23 Wang, T., Chen, C., Larcher, L. M., Barrero, R. A. & Veedu, R. N. Three decades of nucleic acid aptamer technologies: Lessons learned, progress and opportunities on aptamer development. *Biotechnology Advances* **37**, 28-50, doi:<https://doi.org/10.1016/j.biotechadv.2018.11.001> (2019).
- 24 Kyle, S. Affimer Proteins: Theranostics of the Future? *Trends in Biochemical Sciences* **43**, 230-232, doi:10.1016/j.tibs.2018.03.001 (2018).
- 25 Agard, N. J., Prescher, J. A. & Bertozzi, C. R. A Strain-Promoted [3 + 2] Azide-Alkyne Cycloaddition for Covalent Modification of Biomolecules in Living Systems. *Journal of the American Chemical Society* **126**, 15046-15047, doi:10.1021/ja044996f (2004).
- 26 McKay, C. S. & Finn, M. G. Click Chemistry in Complex Mixtures: Bioorthogonal Bioconjugation. *Chemistry & Biology* **21**, 1075-1101, doi:10.1016/j.chembiol.2014.09.002 (2014).
- 27 Hui, C. *et al.* Core-shell Fe₃O₄@SiO₂ nanoparticles synthesized with well-dispersed hydrophilic Fe₃O₄ seeds. *Nanoscale* **3**, 701-705, doi:10.1039/c0nr00497a (2011).
- 28 Ansari, S. A. & Husain, Q. Potential applications of enzymes immobilized on/in nano materials: A review. *Biotechnology Advances* **30**, 512-523, doi:<https://doi.org/10.1016/j.biotechadv.2011.09.005> (2012).
- 29 Kango, S. *et al.* Surface modification of inorganic nanoparticles for development of organic-inorganic nanocomposites—A review. *Progress in Polymer Science* **38**, 1232-1261, doi:<https://doi.org/10.1016/j.progpolymsci.2013.02.003> (2013).
- 30 Zhang, L., He, R. & Gu, H.-C. Oleic acid coating on the monodisperse magnetite nanoparticles. *Applied Surface Science* **253**, 2611-2617, doi:<https://doi.org/10.1016/j.apsusc.2006.05.023> (2006).
- 31 Mueller, R., Kammler, H. K., Wegner, K. & Pratsinis, S. E. OH Surface Density of SiO₂ and TiO₂ by Thermogravimetric Analysis. *Langmuir* **19**, 160-165, doi:10.1021/la025785w (2003).
- 32 Cuatrecasas, P. Protein Purification by Affinity Chromatography: DERIVATIZATIONS OF AGAROSE AND POLYACRYLAMIDE BEADS. *Journal of Biological Chemistry* **245**, 3059-3065 (1970).
- 33 Chang, Y.-H. *et al.* New Mass-Spectrometry-Compatible Degradable Surfactant for Tissue Proteomics. *Journal of Proteome Research* **14**, 1587-1599, doi:10.1021/pr5012679 (2015).
- 34 Brown, K. A. *et al.* A photocleavable surfactant for top-down proteomics. *Nature Methods* **16**, 417-420, doi:10.1038/s41592-019-0391-1 (2019).
- 35 Loo, R. R. O., Dales, N. & Andrews, P. C. Surfactant effects on protein structure examined by electrospray ionization mass spectrometry. *Protein Science* **3**, 1975-1983, doi:10.1002/pro.5560031109 (1994).
- 36 Janes, K. A. Fragile epitopes—Antibody's guess is as good as yours. *Science Signaling* **13**, eaaz8130, doi:10.1126/scisignal.aaz8130 (2020).
- 37 Silveira, J. A., Ridgeway, M. E., Laukien, F. H., Mann, M. & Park, M. A. Parallel accumulation for 100% duty cycle trapped ion mobility-mass spectrometry. *International Journal of Mass Spectrometry* **413**, 168-175, doi:<https://doi.org/10.1016/j.ijms.2016.03.004> (2017).
- 38 Yu, Q. *et al.* Benchmarking the Orbitrap Tribrid Eclipse for Next Generation Multiplexed Proteomics. *Analytical Chemistry* **92**, 6478-6485, doi:10.1021/acs.analchem.9b05685 (2020).
- 39 Toby, T. K., Fornelli, L. & Kelleher, N. L. Progress in Top-Down Proteomics and the Analysis of Proteoforms. *Annual Review of Analytical Chemistry* **9**, 499-519, doi:10.1146/annurev-anchem-071015-041550 (2016).
- 40 Roberts, D. S. *et al.* Reproducible large-scale synthesis of surface silanized nanoparticles as an enabling nanoproteomics platform: Enrichment of the human heart phosphoproteome. *Nano Research* **12**, 1473-1481, doi:10.1007/s12274-019-2418-4 (2019).
- 41 Bates, K. J. *et al.* Circulating Immunoreactive Cardiac Troponin Forms Determined by Gel Filtration Chromatography after Acute Myocardial Infarction. *Clinical Chemistry* **56**, 952, doi:10.1373/clinchem.2009.133546 (2010).

- 42 Wu, A. H. B. *et al.* Characterization of cardiac troponin subunit release into serum after acute myocardial infarction and comparison of assays for troponin T and I. *Clinical Chemistry* **44**, 1198 (1998).
- 43 Labugger, R., Organ, L., Collier, C., Atar, D. & Van Eyk Jennifer, E. Extensive Troponin I and T Modification Detected in Serum From Patients With Acute Myocardial Infarction. *Circulation* **102**, 1221-1226, doi:10.1161/01.cir.102.11.1221 (2000).
- 44 Park, K. C., Gaze, D. C., Collinson, P. O. & Marber, M. S. Cardiac troponins: from myocardial infarction to chronic disease. *Cardiovascular Research* **113**, 1708-1718 (2017).
- 45 McDonough, J. L. & Van Eyk, J. E. Developing the next generation of cardiac markers: Disease-induced modifications of troponin I. *Progress in Cardiovascular Diseases* **47**, 207-216, doi:<https://doi.org/10.1016/j.pcad.2004.07.001> (2004).
- 46 Zhang, J. *et al.* Top-Down Quantitative Proteomics Identified Phosphorylation of Cardiac Troponin I as a Candidate Biomarker for Chronic Heart Failure. *Journal of Proteome Research* **10**, 4054-4065, doi:10.1021/pr2002S8m (2011).
- 47 Lewis Joshua, R. *et al.* Association Between High-Sensitivity Cardiac Troponin I and Cardiac Events in Elderly Women. *Journal of the American Heart Association* **6**, e004174, doi:10.1161/jaha.116.004174 (2017).
- 48 Apple, F. S., Sandoval, Y., Jaffe, A. S. & Ordonez-Llanos, J. Cardiac Troponin Assays: Guide to Understanding Analytical Characteristics and Their Impact on Clinical Care. *Clinical Chemistry* **63**, 73, doi:10.1373/clinchem.2016.255109 (2017).

REVIEWERS' COMMENTS:

Reviewer #1 (Remarks to the Author):

The authors have addressed the reviewer's comments. The reviewer recommends the acceptance of this well-revised manuscript.

Reviewer #2 (Remarks to the Author):

The authors have adequately addressed all the reviewers' comments. Nitpicky detail but for completeness, they should provide experimental details for obtaining top-down tandem mass spectra (Supplementary Figure 29).

Reviewer Comments

Reviewer #1 (Remarks to the Author):

The authors have addressed the reviewer's comments. The reviewer recommends the acceptance of this well-revised manuscript.

Response: We are grateful to the Reviewer for the highly positive comments.

Reviewer #2 (Remarks to the Author):

The authors have adequately addressed all the reviewers' comments. Nitpicky detail but for completeness, they should provide experimental details for obtaining top-down tandem mass spectra (Supplementary Figure 29).

Response: We thank the Reviewer for the constructive comments. Following the Reviewer's suggestion, we have added experimental details in the Methods for obtaining top-down tandem mass spectra (new Supplementary Figure 30): "For the targeted collision-induced dissociation (CID) LC-MS/MS analysis shown in Supplementary Fig. 30, the collision energy was varied from 18V to 30V and the quadrupole low mass was set to 500 m/z with a scan range of 200 to 3000 m/z . The total ion current (TIC) corresponding to all obtained MS/MS signal was averaged across the LC retention window corresponding to cTnI, and the averaged MS/MS data was then directly imported into MASH Explorer for proteoform identification and sequence mapping."